# A Novel and Efficient Phthalate Hydrolase from *Acinetobacter* sp. LUNF3: Molecular Cloning, Characterization and Catalytic Mechanism

**DOI:** 10.3390/molecules28186738

**Published:** 2023-09-21

**Authors:** Shuanghu Fan, Jingjing Guo, Shaoyan Han, Haina Du, Zimeng Wang, Yajuan Fu, Hui Han, Xiaoqiang Hou, Weixuan Wang

**Affiliations:** 1College of Life Science, Langfang Normal University, Langfang 065000, China; fanshuanghu@126.com (S.F.); hanshaoyan2023@163.com (S.H.); 18731308321@163.com (H.D.); wzm20030512@163.com (Z.W.); fuyajuan501@163.com (Y.F.); hh071@163.com (H.H.); 2Institute of Agricultural Resources and Regional Planning, Chinese Academy of Agricultural Sciences, Beijing 100081, China; 3Technical Innovation Center for Utilization of Edible and Medicinal Fungi in Hebei Province, Langfang 065000, China; 4School of Chemistry and Materials Science, Langfang Normal University, Langfang 065000, China; guojingjing430@163.com; 5Biotechnology Research Institute, Chinese Academy of Agricultural Sciences, Beijing 100081, China; 6National Nanfan Research Institute (Sanya), Chinese Academy of Agricultural Sciences, Sanya 572024, China

**Keywords:** *Acinetobacter*, biodegradation, genome sequencing, HSL family hydrolase, site-directed mutagenesis

## Abstract

Phthalic acid esters (PAEs), which are widespread environmental contaminants, can be efficiently biodegraded, mediated by enzymes such as hydrolases. Despite great advances in the characterization of PAE hydrolases, which are the most important enzymes in the process of PAE degradation, their molecular catalytic mechanism has rarely been systematically investigated. *Acinetobacter* sp. LUNF3, which was isolated from contaminated soil in this study, demonstrated excellent PAE degradation at 30 °C and pH 5.0–11.0. After sequencing and annotating the complete genome, the gene *dphAN1*, encoding a novel putative PAE hydrolase, was identified with the conserved motifs catalytic triad (Ser^201^-Asp^295^-His^325^) and oxyanion hole (H^127^GGG^130^). DphAN1 can hydrolyze DEP (diethyl phthalate), DBP (dibutyl phthalate) and BBP (benzyl butyl phthalate). The high activity of DphAN1 was observed under a wide range of temperature (10–40 °C) and pH (6.0–9.0). Moreover, the metal ions (Fe^2+^, Mn^2+^, Cr^2+^ and Fe^3+^) and surfactant TritonX-100 significantly activated DphAN1, indicating a high adaptability and tolerance of DphAN1 to these chemicals. Molecular docking revealed the catalytic triad, oxyanion hole and other residues involved in binding DBP. The mutation of these residues reduced the activity of DphAN1, confirming their interaction with DBP. These results shed light on the catalytic mechanism of DphAN1 and may contribute to protein structural modification to improve catalytic efficiency in environment remediation.

## 1. Introduction

Phthalic acid esters (PAEs) have been used as plasticizers since the 1920s, improving the flexibility and durability of plastic products [1,2]. However, PAEs are likely to be released into the environment from plastics due to their non-covalent interactions with plastic matrices [3]. Currently, the serious environmental contamination caused by PAEs is leading to increasing concern among researchers worldwide. Moreover, PAEs have similar properties to endocrine disruptors, carcinogens and mutagens [4], posing a potential threat to humans and ecological environments [5,6]. Given the environmental risk, six kinds of PAEs have been listed as priority environmental pollutants: dimethyl phthalate (DMP), diethyl phthalate (DEP), dibutyl phthalate (DBP), benzyl butyl phthalate (BBP), dioctyl phthalate (DOP) and di(2-ethylhexyl) phthalate (DEHP). In view of the serious pollution and toxicological effects of PAEs, an efficient strategy for remediating PAE-contaminated environments must be urgently developed.

Due to their high hydrophobicity, it is difficult for PAEs to be naturally degraded via hydrolysis and photolysis [7,8]. Biodegradation via microorganisms has become the main approach to eliminate PAEs from environments with the advantages of high efficiency and environmental safety. Numerous PAE-degrading bacteria have been isolated from environmental matrices and mainly belong to the genera *Sphingomonas* [9], *Gordonia* [10], *Rhodococcus* [11], *Arthrobacter* [12], *Bacillus* [13] and *Pseudomonas* [14]. These bacteria degrade PAEs mainly via aerobic metabolism, and the degradation characteristics of multiple strains have been revealed. *Rhodococcus pyridinivorans* XB isolated from activated sludge has a high degradation efficiency towards PAEs, removing 98% of DMP, DEP and DBP within 5 days [11]. In some cases, some functional strains work together to completely degrade PAEs [15]. Some strains or consortiums have also been applied to PAE remediation in various environments. *Gordonia* sp. QH-11 and *Gordonia* sp. Lff can efficiently remove PAEs from soil and alter the composition of the bacterial community in soil [10,16].

Traditionally, the initial hydrolysis of PAEs is considered the most important step during the whole degradation process, and causal hydrolases are considered the key enzymes during the metabolism of PAEs via bacteria. The dialkyl PAE hydrolase is responsible for the hydrolysis of the first ester bond of PAEs [17] (Figure 1). The resulting monoalkyl PAEs, with only one ester bond, are transformed into phthalic acid (PA) by monoalkyl PAE hydrolase [18]. Occasionally, the same hydrolase is able to function in the two steps of hydrolysis, such as EstG, GTW28_17760 and EstM2 from *Sphingobium* sp. SM42, *Bacillus subtilis* BJQ0005 and a soil metagenomic library, respectively [19,20,21]. Some dialkyl PAE hydrolases have been reported, but their gene sequences remain unidentified. The hydrolase purified from the culture broth of *Nocardia erythropolis* not only hydrolyzes several kinds of PAEs, including DBP, DEHP, DEP, DOP, etc., but also hydrolyzes dimethyl isophthalate (DMIP), dimethyl terephthalate (DMTP) and diethyl terephthalate (DETP) [22]. The DMP hydrolases from the cell-free extract of *Bacillus* sp. are characterized, and their genes may be located in the plasmid [23]. Fortunately, metagenomic library, bacterial genomic library and genome sequencing provide effective approaches to identifying gene sequences of dialkyl PAE hydrolases such as Hyd [24], XtjR8 [25], HylD1 [26] and EstYZ5 [17]. These enzymes mainly belong to hydrolase families II, IV, V, VI, VII and VIII [20,21,26] (Table 1). Among them, the hydrolases from family IV, also known as the hormone-sensitive lipase (HSL) family, predominate. These hydrolases from the HSL family possess conserved motifs, such as oxygen hole (HGGG), catalytic triad (Ser-Asp/Glu-His) and GDSAG containing the catalytic Ser [26]. However, the relevant studies of PAE hydrolases mainly focus on gene cloning, function verification, catalytic characteristics and structure analysis. Although the catalytic mechanisms of dialkyl PAE hydrolase PS06828 and Hyd, from family VI and a new family, respectively, have been inferred from spectroscopic and docking analyses [24,27], only the catalytic triad of PS06828 (Ser113, Asp166 and His197) and key residues of Hyd (Thr190 and Ser191) are revealed. According to the results of molecular docking and enzyme assay, the interaction between MIBP and His399 affects the activity of GTW28_17760, a hydrolase from family VII capable of hydrolyzing DIBP and MIBP [20]. The catalytic mechanisms of some monoalkyl PAE hydrolases from family V are also elucidated [18,28]. Altogether, the catalytic mechanisms of PAE hydrolases from other families require to be further resolved.

This study aimed to explore PAE-degrading bacteria, characterize the degradation performance of the isolate, identify efficient PAE hydrolases and elucidate their molecular catalytic mechanisms. The complete genome of *Acinetobacter* sp. LUNF3, an effective PAE degrader, was sequenced and annotated. The novel dialkyl PAE hydrolase DphAN1 was identified via genomic analysis. After the heterologous expression and affinity purification of DphAN1, its catalytic characteristics were determined. DphAN1 is endowed with a high activity towards PAEs, particularly recalcitrant BBP, and has a good adaptability to some chemicals. The interaction between DphAN1 and DBP via molecular docking revealed active residues related to binding and hydrolyzing DBP. These active residues were validated by site-directed mutagenesis and an enzyme assay. The results provide insights into the molecular catalytic mechanism of DphAN1 and will support its structural modification, aiming for a higher catalytic efficiency.

## 2. Results and Discussion

### 2.1. Isolation and Identification of Acinetobacter sp. LUNF3

After enrichment using DBP as a substrate, the DBP-degrading ability of some candidate strains was assessed by HPLC-MS. One strain, LUNF3, significantly removed DBP and produced MBP (Appendix A and Figure 2A), indicating its efficiently hydrolyzing DBP. The colony of strain LUNF3 is light yellow with a circular and smooth surface. The 16S rRNA gene of strain LUNF3 shares the maximum sequence identity to that of *Acinetobacter* sp. based on BLAST against NCBI database (Appendix A). According to the phylogenetic tree, strain LUNF3 is taxonomically close to strains *Acinetobacter johnsonii* CIP 64.6 and *Acinetobacter oryzae* B23 from EzBioCloud database (Figure 2B). Thus, the DBP-degrading strain here is identified as *Acinetobacter* sp. LUNF3. It was reported that some strains of the genus *Acinetobacter* were able to degrade PAEs [29,30,31]. *Acinetobacter* sp. LMB-5 isolated from soil can quickly remove DMP, DEP and DBP [30]. *Acinetobacter* sp. SN13 is capable of degrading DEHP [31]. The isolation of *Acinetobacter* sp. LUNF3 enriches the resources of PAE-degrading bacteria. The genus *Acinetobacter* can also eliminate other pollutants, such as sulfamethoxazole [32], perfluorooctane sulfonamide [33], crude oil [34] and phenol [35]. Therefore, they have a potential application in the remediation of contaminated environments.

### 2.2. Characterization of PAE Degradation by Strain LUNF3

Environmental parameters, such as temperature and pH, affect PAE degradation due to their influence on the enzymatic activity of microorganisms. Strain LUNF3 exhibits the highest degradation efficiency at 30 °C, with the complete elimination of DBP (Figure 3A). Approximately 30% of DBP is degraded even at 10 °C, indicating the relatively high activity of LUNF3 towards DBP at low temperature. In addition, strain LUNF3 can effectively remove DBP at a wide range of pH values. The degradation percentage of DBP greatly increases with an increase in pH, rising from 4.0 to 9.0, and is maintained at 100% until pH reached 11.0 (Figure 3B). Strain LUNF3 eliminates around 70% of DBP at pH 5.0 and 6.0. Generally, microorganisms cannot effectively degrade PAEs at a pH too low or too high. *Enterobacter* sp. DNB-S2 completely degrades DBP at pH 7.0–10.0 but only removes 4.4% and 10.4% of DBP at pH 5.0 and 6.0, respectively [36]. *Rhodococcus pyridinivorans* DNHP-S2 degrades more than 40% of DEHP at pH 5.0 [37]. The natural environmental conditions are complex and diverse. The high degradation efficiency of strain LUNF3 indicates its promising practical application in environmental remediation across a broad pH range, especially in neutral or alkaline environments (Figure 3A,B).

In a substrate profile test, strain LUNF3 mediates degradation of more than 70% of DEP, as well as 100% of DBP and BBP (Appendix A). The degradative activity of strain LUNF3 against PAEs varies with their different side chains. Particularly, strain LUNF3 effectively degrades BBP despite the potential steric hindrance by its bulky side chain [38,39]. Based on the HPLC-MS profiles of DEP and BBP, which were degraded by strain LUNF3 (Appendix A and Figure 3C), the products of MEP, MBP and MBeP were identified based on retention time and molecular ions. The results indicate that strain LUNF3 degrades these kinds of PAEs via hydrolysis. Generally, hydrolysis is a key step for PAE degradation and detoxification [40]. The bacterial degradation of PAEs mainly begins with hydrolysis, such as *Burkholderia pyrrocinia* B1213 [41], *Achromobacter* sp. RX [42], *Rhodococcus pyridinovorans* DNHP-S2 [37] and *Pseudarthrobacter defluvii* E5 [43]. Only a few bacteria adopt β-oxidation to initiate PAE degradation [15,44].

### 2.3. Cloning of PAE Hydrolase Gene through Complete Genome Sequencing

Strain LUNF3 underwent complete genome sequencing in order to explore the candidate genes involved in PAE degradation. PacBio sequencing produced 281,192 reads with 2,095,890,875 bp (Appendix A), which were applied to genome assembly. Illumina sequencing generated high-quality data of 4,840,844 reads with 1,188,882,811 bp, which were used to correct the assembled genome. The obtained complete genome comprises a chromosome with 3,317,686 bp (a GC content of 41.43%) and a circular plasmid with 101,829 bp (a GC content of 38.14%) (Appendix A, Figure 4). The genome size and GC content of strain LUNF3 are similar to those of *Acinetobacter johnsonii* LXL_C1 and *Acinetobacter* sp. TTH0–4 [45,46]. *Acinetobacter* sp. TTH0-4 can degrade crude oil and has a chromosome of 2,962,453 bp and 38.74% GC. The cyprodinil degrader *Acinetobacter johnsonii* LXL_C1 harbors a chromosome of 3,398,706 bp with 41.2% GC content. Based on comparative genomic analysis, the average nucleotide identity (ANI) values of strain LUNF3 against *Acinetobacter johnsonii* LXL_C1 (GenBank No. CP031011), *Acinetobacter johnsonii* M19 (GenBank No. CP037424) and *Acinetobacter johnsonii* FDAARGOS_1093 (GenBank No. CP068195) were 97.75%, 96.17% and 95.85%, respectively. Therefore, strain LUNF3 is genetically close to *Acinetobacter johnsonii* and can be a member of this bacterial species.

Based on the genome annotation, the chromosome and plasmid of strain LUNF3 harbor 3177 and 98 protein-coding genes (CDSs), respectively (Table 1). Moreover, all seven copies of rRNA genes (5S, 16S, 23S), eighty-nine tRNA genes and one CRISPR structure are located in the chromosome. Among these CDSs, 3221, 2162 and 2270 genes are annotated in NR, Swiss-Prot and GO database (Appendix A, Appendix A), respectively. There are 1607 genes distributing across 46 biological pathways in the KEGG database, with 65 genes related to biodegradation and metabolism of xenobiotics (Appendix A). A total of 2759 genes are divided into 20 function categories according to annotations in the COG database (Appendix A), and 109 genes are involved in lipid transport and metabolism. PAE degradation by bacteria is mediated by metabolism enzymes such as hydrolases, which are involved in cleaving ester bonds of PAEs, the critical step during PAE degradation [1,47,48]. Among these 109 genes mentioned above, one gene encodes a hydrolase DphAN1, which shares 67.0% identity with reported PAE hydrolase from *Acinetobacter* sp. M673 [29]. The result implies putative PAE-hydrolyzing capability of DphAN1.

According to the phylogenetic relationship between DphAN1 and the hydrolases from families I-VIII, DphAN1 was classified into family IV, which included the reported PAE hydrolases EstS1(AEW03609.1) [49], DphB (AGY55960.1) [50], EstG (AJO67804.1) [19] and the hydrolase (AFK31309.1) from *Acinetobacter* sp. M673 [29] (Figure 5A). Thus far, the molecular cloning and enzymatic characterization of some bacterial PAE hydrolases, which belong to families II, IV, V, VI, VII and VIII, have been conducted [20,21,26]. Furthermore, after mining the transcriptome of *Cylindrotheca closterium* using known bacterial PAE hydrolases as probes, two putative DBP hydrolases, DBPH1 and DBPH2, were predicted from the diatom for the first time [51]. The interaction of DBPH1 and DBPH2 with DBP and their expression levels under DBP exposure highlight their potential PAE-hydrolyzing ability. The conserved motifs of DphAN1 were analyzed using multiple sequence alignment. The typical catalytic triad (Ser^201^-Asp^295^-His^325^) was detected in DphAN1 (Appendix A). EstJ6 and EstYZ5, the PAE hydrolases from family IV, possess Glu instead of Asp in the catalytic triad [17,48]. The conserved G^199^DSAGG^204^ is also present in DphAN1 and contains the Ser^201^ of the catalytic triad (Figure 5B). In DphB, a cold-active PAE hydrolase identified from the metagenomic library of biofilms, Glu replaces Asp of the GDSAGG motif [50]. The motif H^127^GGG^130^ is also conserved (Figure 5C) and may form an oxyanion hole to stabilize reaction intermediates during the catalyzation of PAEs via DphAN1. Other conserved motifs were also observed in DphAN1, including D^295^LLHDEG^301^ containing the catalytic Asp^295^, H^325^GF^327^ harboring the catalytic His^325^, and V^154^LSIDYPLAPE^164^ (Appendix A). The functions of these conserved residues are further confirmed in this study.

### 2.4. Functional Identification and Characterization of DphAN1

To identify the enzymatic activity of DphAN1 toward PAEs, the gene *dphAN1* was expressed in ArcticExpress (DE3) cells. After purification via affinity chromatography, the recombinant DphAN1 with 6 × His tag was analyzed using SDS-PAGE. One single band was observed with a molecular weight of approximately 40 kDa (Figure 6A), in accordance with the theoretically calculated value of recombinant DphAN1 (41.46 kDa). The high PAE-hydrolyzing activity of recombinant DphAN1 was detected. After treatment with DphAN1, 51.34%, 77.02% and 58.60% of DEP, BBP and DBP were eliminated with corresponding enzymatic specific activity of 1.81, 2.72 and 2.07 U/mg protein (Figure 6B), respectively. DphAN1 catalyzed the hydrolysis of PAEs and generated the corresponding monoalkyl PAEs according to HPLC-MS (Figure 6C and Appendix A). Importantly, the hydrolysis of BBP generates MBP and MBeP, suggesting that DphAN1 is capable of cleaving either of the two esters of BBP (Appendix A).

Environmental factors have a critical impact on enzymatic activity [52]. So, it is necessary to explore the optimal conditions for enzymatic catalysis. The enzyme assay of DphAN1 was investigated under various environmental conditions. The highest activity was observed at pH 8.0, and more than 57% of activity was maintained at pH 6.0–9.0 (Figure 7A). EstYZ5 displays less than 10% of the highest activity at pH 5.0 [17]. The activity of EstM2 is not reported at pH < 6.0 [21]. Therefore, DphAN1 has the advantage of high activity in slightly acid to alkaline environments. The high activity of DphAN1 remains between 20 °C and 40 °C, with the optimum activity at 30 °C (Figure 7B). Notably, DphAN1 displays 47.08% of maximal activity even at 10 °C. Some PAE hydrolases show relatively low activity at 10 °C in contrast to those at optimal temperatures, such as Est3563 and EstS1 [49,53]. The better environmental adaptability of DphAN1 will make possible its practical application in environments with a wide range of pH and temperature (Figure 7A,B).

The effects of chemicals on the activity of DphAN1 were also investigated. The metal ions at a concentration of 1 mM positively or negatively affect the activity of DphAN1 to different extents (Figure 7C). The addition of Mn^2+^, Cr^2+^, Fe^2+^ and Fe^3+^ causes an increase in the activity of DphAN1 by 18–89%. In a previous report, Fe^3+^ also enhances oxytetracycline degradation by *Pseudomonas* sp. T4 [54]. DphAN1 is not very sensitive to Ca^2+^, Mg^2+^ and Cd^2+^, indicating the relative stability of DphAN1 under these ions. However, Co^2+^, Ni^2+^, Zn^2+^ and Cu^2+^ reduce its activity by 20–40%, while Hg^2+^ inactivates DphAN1 completely. A significant decrease in activity was observed under 1% of Tween20 and Tween80, while the activity was lost when the concentration of Tween20 and Tween80 reached 5% (Figure 7D). These results demonstrate that the inhibitory effects were gradually enhanced with the increased concentration of Tween20 and Tween80. The strong denaturant SDS brings about a complete loss of activity, which may be caused by the destruction of DphAN1 by SDS. The surfactant Triton X-100 serves as an activator of DphAN1 and leads to an increase in activity by 49%. The Ser-specific inhibitor PMSF and His modifier DEPC cause complete inactivation of DphAN1. These findings suggest that the residues Ser and His are involved in catalysis, which is consistent with the components of the catalytic triad (Ser^201^-Asp^295^-His^325^). Phenyl methane sulfonyl fluoride (PMSF) and diethylpyrocarbonate (DEPC) possibly modify Ser^201^ and His^325^, respectively, and ultimately influence the hydrolytic activity. More than 60% of activity is maintained after the addition of β-mercaptoethanol (β-ME). Three Cys residues of DphAN1 may form disulfide bonds, which are disrupted under the action of β-ME, thus reducing the activity.

### 2.5. Structural Analysis of DphAN1

The three-dimensional structure of DphAN1 was modeled with the template of Est8 (PDB:4ypv) [55] (Figure 8A). According to validation by Molprobity [56], 94.8% of residues are in favored (98%) regions and 98.7% in allowed (>99.8%) regions (Appendix A), demonstrating the good quality of the modeled DphAN1 and its suitability for future research. The superposition of the structures of DphAN1 and Est8 resulted in an RMSD value of 0.149 (Figure 8B), indicating that DphAN1 belongs to the a/β hydrolase superfamily and harbors catalytic domain and cap domain (Figure 8A,B). In the catalytic domain, the central mixed β-sheet is sandwiched by several α-helices. The putative catalytic triad (Ser^201^-Asp^295^-His^325^) and oxyanion hole components H^127^GGG^130^ are situated in flexible loops or coils of the catalytic domain (Figure 8A), implying the location of the substrate-binding pocket. The residue Ser^201^ was in proximity with the motif H^127^GGG^130^ in accordance with their related functions. The orientation of the catalytic triad and oxyanion hole of DphAN1 may facilitate its interacting and hydrolyzing PAEs. The cap domain is composed of five α-helices and might determine the entrance of PAEs to the substrate-binding pocket [55].

Generally, molecular docking serves as an effective method for evaluating the interaction between the protein and ligand [25]. Molecular docking analysis of DphAN1 and DBP was performed to explore the molecular basis of DphAN1 catalyzing DBP hydrolysis (Figure 8C). DBP is located in the area between the catalytic domain and cap domain (Figure 8C), which is a long and hydrophobic pocket (Figure 8D and Appendix A). The catalytic Ser^201^ was situated around the substrate-binding pocket. The hydroxyl oxygen of Ser^201^ is 3.5 Å away from the carboxyl carbon of DBP, which could facilitate nucleophilic attack of Ser^201^ against the carboxyl carbon of DBP like other hydrolases [18]. In addition, the interactions among Ser^201^, His^325^ and Asp^295^ in the putative catalytic triad are mediated by hydrogen bonds, which might participate in electron transport during the process of DphAN1 catalyzing DBP (Figure 8D). To validate the deduction, Ser^201^, His^325^ and Asp^295^ were replaced by Ala via site-directed mutagenesis, respectively (Appendix A). After the enzyme assay, all variants displayed no hydrolytic activity towards DBP (Figure 8E), confirming that these amino acids are components of the catalytic triad. The catalytic triads are also identified in PAE hydrolases Xtj8 (Ser^152^-Glu^246^-His^276^) and EstJ6 (Ser^146^-Glu^240^-His^270^) from the HSL family [25,48], and the residue Glu substitutes Asp of the catalytic triad. The backbone amide (NH) of Gly^129^ and Gly^130^ forms an oxyanion hole (Figure 8D) and may stabilize the reaction tetrahedral intermediate of DphAN1-DBP during a hydrolysis process similar to that observed in other hydrolases [57,58,59]. Although the oxyanion hole is a common feature of hydrolases [57,59], the constituent residues are of diversity among hydrolases, such as Phe^71^ and Met^176^ of Est22 [60], Cys^40^ and Ser^120^ of AlAXEA [61], and Tyr^87^ and Met^160^ of IsPETase [57]. The variants G129A and G130A exhibited approximately 40% hydrolytic activity compared to the wild type of DphAN1 (Figure 8D), implying their key roles in the oxyanion hole. Moreover, some other residues of DphAN1 are involved in interaction with DBP to enhance the substrate-binding force (Figure 8D). The phenyl group of DBP forms π-π stack with Phe^78^ and π-anion interaction with Asp^253^, respectively. The residues Val^133^, Val^254^, Val^230^, Val^257^ and Phe^330^ provide a relatively hydrophobic pocket to stably bind to DBP (Figure 8D and Appendix A). When these residues were mutated to Ala, the activity of variants was significantly reduced. The possible reason is the mutation causes weak interaction between the enzyme and DBP. Based on the reduced activity of DphAN1 variants, the active residues responsible for binding and hydrolyzing DBP are identified in DphAN1 and can facilitate structural modification by rational design to improve catalytic performance [58].

## 3. Materials and Methods

### 3.1. Reagents and Media

PAEs and their metabolites were obtained from Aladdin Chemistry Co., Ltd. (Shanghai, China) or Sinopharm Chemical Reagent Co., Ltd. (Shanghai, China) and included the following: DEP, DBP, BBP, monoethyl phthalate (MEP), monobutyl phthalate (MBP), monobenzyl phthalate (MBeP) and phthalic acid (PA). Methanol was of HPLC grade, and other chemicals were of analytical grade. PAEs were dissolved in methanol to prepare the stock solution. The restriction enzymes, ligase, TA/Blunt-Zero Cloning vector, Mut Express II and Ni-NTA Resin were purchased from Vazyme Biotech Co., Ltd. (Nanjing, China) or TransGen Biotech Co., Ltd. (Beijing, China). The binding buffer and elution buffer were prepared using our existing methods [59]. The media for bacteria culture included Luria-Bertani (LB) medium and trace element medium (TEM) [62].

### 3.2. Enrichment, Isolation and Identification of PAE-Degrading Bacteria

Soil was collected from a wasteyard in Xingtai, China. The sample (10 g) was added to 100 mL of liquid LB medium containing 0.5 mM DBP and incubated at 180 rpm and 30 °C for 5 days. Then, 1% of the culture was transferred to fresh TEM (pH 8.0) with 0.5 mM DBP for further cultivation under the same conditions. The enrichment step was repeated five times in TEM (pH 8.0). The culture was streaked on a TEM agar plate with 0.5 mM DBP and incubated at 30 °C for 5 days. Some colonies, potentially capable of degrading DBP, were incubated in 100 mL of LB liquid medium to OD600 = 1.0, respectively. The cells were centrifuged, washed and resuspended in 200 mL of TEM (pH 8.0). The suspension was divided into 10 mL, supplemented with 0.5 mM DBP and cultivated for 5 days at 180 rpm and 30 °C. The control contained 10 mL of TEM (pH 8.0) with 0.5 mM DBP. All experiments were conducted in triplicate in the present study. The degradative activity of strain LUNF3 was estimated by HPLC-MS assay of DBP hydrolysis.

The genomic DNA of DBP-degrading strain LUNF3 was prepared using EasyPure^®^ Bacteria Genomic DNA Kit (TransGen, Beijing, China). The DNA was used as template for amplification of 16S rRNA gene with primers 27F and 1492R [37]. The amplified products were ligated with TA/Blunt-Zero Cloning vector (Vazyme, Nanjing, China), which were transferred into DH5α. After sequencing and assembly of the 16S rRNA gene, the gene sequence was submitted in NCBI (https://blast.ncbi.nlm.nih.gov/Blast.cgi?PROGRAM=blastn&PAGE_TYPE=BlastSearch&LINK_LOC=blasthome (accessed on 1 April 2022)) and EzBioCloud Database (https://www.ezbiocloud.net/ (accessed on 1 April 2022)) to search for the relatives. The phylogenetic tree was constructed via MEGA 6.0 using the neighbor-joining method.

### 3.3. Degradation of PAEs by Strain LUNF3

The suspension of strain LUNF3 was prepared in liquid TEM (pH 8.0) with 0.5 mM DBP, as mentioned above. To investigate DBP degradation performance of the isolated strain at a range of temperatures (20, 30, 40 and 50 °C), the cultures were incubated at 180 rpm and a certain temperature for 5 days. To examine the effect of pH on DBP degradation, the strain LUNF3 was suspended in TEM (0.5 mM DBP) at a range of pH 4.0–11.0, and other culture conditions were set as above. The TEM (pH 8.0) with 0.5 mM DBP was adopted as the control. The concentration of residual DBP was determined by HPLC. To investigate the substrate spectra of the isolated strain, 0.5 mM of PAEs (DEP, DBP and BBP individually) were added into the strain suspension in TEM (pH 8.0). The mixtures were cultivated at 180 rpm, pH 8.0 and 30 °C for 5 days. The concentration of residual PAEs was determined using HPLC, and the degradation metabolites were detected by HPLC-MS [63].

### 3.4. Complete Genome Sequencing and Annotation

The genomic DNA of strain LUNF3 was prepared using EasyPure^®^Bacteria Genomic DNA Kit (TransGen Biotech, Beijing, China) according to the manufacturer’s instructions. The integrity and purity were detected via agarose gel electrophoresis and Qubit Fluorometer, respectively. After the construction of genome sequencing libraries, sequencing was carried out on the Illumina NovaSeq platform and PacBio Sequel platform. The genome sequence was assembled using PacBio sequencing reads via HGAP and CANU [64,65] and rectified using sequencing results from the Illumina platform. Based on the complete genome sequence, protein-coding genes, tRNA genes and rRNA genes were predicted using GeneMarkS, Trnascan-se and RNAmmer [66,67,68], respectively. The genes were annotated in public databases such as NR, eggNOG, KEGG, Swiss-Prot and GO [10]. The average nucleotide identity (ANI) analysis of the genomes was performed using FastANI [11].

### 3.5. Sequence Analysis and Expression of PAE Hydrolase DphAN1

The genome of strain LUNF3 was mined and searched for similar sequences of PAE hydrolases reported, such as XtjR8, HylD1, HylD2 and EstM2 [21,25,26]. The phylogenetic analysis of predicted PAE hydrolase DphAN1 and lipolytic enzymes from families I-VIII was performed using MEGA 6.0 using the neighbor-joining method. Multiple sequence alignment was conducted using CLUSTALW (https://www.ebi.ac.uk/Tools/msa/clustalo/ (accessed on 1 April 2022), and the conserved sequences were analyzed. After the codon optimization of gene *dphAN1*, the gene was synthesized and cloned into pCold II at the restriction sites of *Nde*I and *Xba*I. The recombinant expression vector pC-*dphAN1* was introduced into strain ArcticExpress (DE3), and the strain was incubated in LB medium with 50 μg/mL ampicillin at 37 °C and 180 rpm. Then, 0.1 mM IPTG was added into the culture when OD_600_ of the culture reached approximately 0.8, inducing the expression of *dphAN1*. After induction of *dphAN1* expression at 16 °C for 20h, the cells were collected by centrifugation and cell disruption was conducted by ultrasonication. The lysate underwent centrifugation to collect the supernatant, which was used to purify recombinant DphAN1 via the column of Ni-NTA Resin (TransGen, Beijing, China) [62]. The concentration of purified DphAN1 was measured using the BCA Protein Assay Kit (TIANGEN, Beijing, China).

### 3.6. Biochemical Characterization of DphAN1

To verify the PAE-hydrolyzing capability of DphAN1, DEP, DBP and BBP were adopted as substrates for tests. For the assays, 0.5 mM substrate and 6.38 μg DphAN1 were added into 0.9 mL of Tris–HCl (50 mM, pH 8.0). The reaction was carried out at 30 °C and 180 rpm for 20 min, and the reaction was terminated by adding 10 µL of 1M HCl. PAEs and hydrolysis products were detected by HPLC or HPLC-MS. The control contained no hydrolase DphAN1. One unit of specific activity of DphAN1 was defined as the amount of enzyme required for hydrolysis of 1 µmol of PAEs per minute. To examine the optimal pH for DphAN1 catalysis, the following buffer Na_2_HPO_4_–citric acid (pH 4.0–7.0), Tris–HCl (pH 8.0–8.9) and Glycine–NaOH (pH 9.0–10.0) was used as the reaction buffer. To examine the optimal temperature, the reaction was performed in Tris–HCl (pH 8.0) at temperatures ranging from 10 °C to 50 °C. Other reaction conditions were set as above. The highest activity under a certain pH or temperature was defined as 100%, and the relative activity under other conditions was calculated.

To investigate the effects of metal ions or chemicals on the activity of DphAN1,these additives, which included metal ions (1 mM Ca^2+^, Mg^2+^, Fe^3+^, Mn^2+^, Zn^2+^, Co^2+^, Ni^2+^, Cu^2+^, Cr^2+^, Cd^2+^, Fe^2+^ and Hg^2+^), surfactants (1 mM and 5 mM SDS, 1% and 5% Tween 20, 1% and 5% Tween 80 and 1% Triton X-100) and esterase inhibitors (1% DEPC, PMSF and β-mercaptoethanol (β-ME)), were individually added into Na_2_HPO_4_–citric acid (pH 7.0). Other reaction conditions were set as above. The activity of DphAN1 was expressed as relative activity to the control without any additives.

### 3.7. Homology Modeling and Molecular Docking

To construct the homology model of DphAN1, the PDB database was searched for homolog proteins using NCBI BLAST. The protein Est8 (PDB:4ypv), an alkaline esterase from a diesel oil-degrading consortium [55], was adopted as a model template due to 34.09% amino acid identity with DphAN1 (over 86% coverage). After homology modeling of DphAN1 using the Modeller software, the modeled structure was checked by Molprobity [56]. The molecular docking of DphAN1 and DBP was performed using AutoDock4.2 [18]. The three-dimensional structure of DBP was obtained from PubChem (http://pubchem.ncbi.nlm.nih.gov/ (accessed on 22 February 2022), and the rotations and torsions of DBP were automatically set in AutoDock tools.

### 3.8. Site-Directed Mutagenesis of DphAN1

The variants of DphAN1 were prepared by site-directed mutagenesis according to the instruction of the kit Mut Express II (Vazyme, Nanjing, China). The residues of DphAN1 (S201, D295, H325, G129, G130, F78, V133, V230, D253, V254, V257 and F330) were replaced by alanine. The mutant primers in Appendix A were applied to amplify the expression vector pC-*dphAN1*. The products were cyclized and transformed into ArcticExpress(DE3) cells. The mutated DphAN1 was overexpressed in these cells under induction by 0.1 mM IPTG. The purification and enzyme assay of variants of DphAN1 were conducted following the procedure described above.

### 3.9. Analytical Methods

PAEs or intermediates in the cultures of strain LUNF3 or reaction mixture of hydrolase DphAN1 were detected by HPLC or HPLC-MS. To extract PAEs or intermediates in the mixture above, an equal volume of ethyl acetate was added with strong agitation. Ethyl acetate in the organic phase was evaporated, and the extracts were dissolved in an equal volume of methanol. The solution was filtrated through a 0.22 µm membrane and then analyzed using HPLC (Agilent 1200, Santa Clara, CA, USA) or HPLC-MS equipped with a triple quadrupole mass spectrometer (Agilent 6420, Santa Clara, CA, USA). The chromatographic conditions were as follows: reverse phase column ZORBAX Eclipse Plus C18 (4.6 mm × 250 mm, 5 µm), the mobile phase of 90% methanol and 10% water containing 0.1% acetic acid, and the flow rate of 0.8 mL/min or 0.5 mL/min.

## 4. Conclusions

In summary, an efficient PAE degrader *Acinetobacter* sp. LUNF3 was isolated from soil under a wasteyard. According to the complete genome sequencing and genomic analysis of strain LUNF3, a novel PAE hydrolase DphAN1 from the HSL family was identified. DphAN1 possesses canonical α/β hydrolase structure and conserved motifs such as catalytic triad and oxyanion hole. DphAN1 is able to hydrolyze numerous kinds of PAEs, especially BBP with bulky side chains. Moreover, DphAN1 displays high activity under a broad range of temperatures and pH values and can tolerate several metal ions and chemicals. The identification and characterization of DphAN1 enriches the resources of PAE hydrolases and significantly highlights its potential application in the remediation of PAE-contaminated environments. The interaction between DphAN1 and DBP was investigated by molecular docking to ascertain key active sites (catalytic triad and other binding residues) and was validated by mutation of these residues. These results provide insights into the catalytic mechanism of hydrolase DphAN1 and may be a basis for protein structural modification to improve catalytic efficiency in environment remediation.

## Figures and Tables

**Figure 1 molecules-28-06738-f001:**
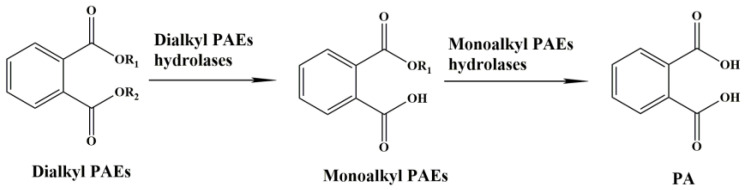
The pathways of PAE hydrolysis by microorganisms.

**Figure 2 molecules-28-06738-f002:**
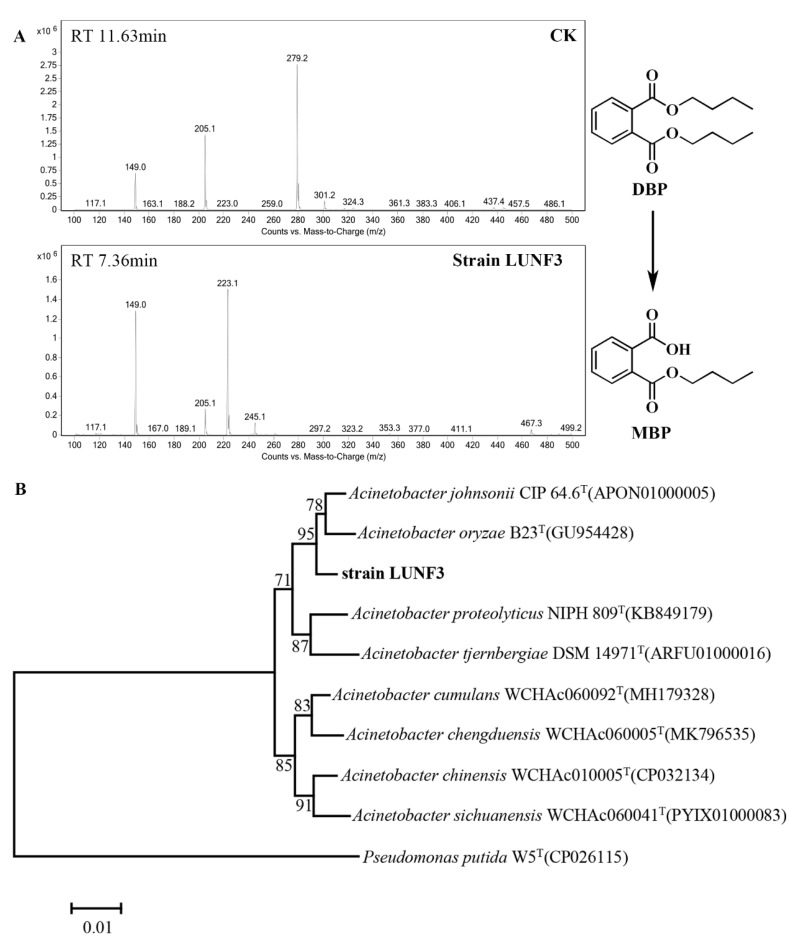
Functional and molecular identification of strain LUNF3. (**A**) HPLC-MS profile of DBP degraded by strain LUNF3 with the metabolite of MBP. Strain LUNF3 was cultivated in TEM (pH 8.0) with 0.5 mM DBP for 5 days at 180 rpm and 30 °C. CK: TEM (pH 8.0) with 0.5 mM DBP. (**B**) The phylogenetic tree constructed using 16S rRNA gene sequences of strain LUNF3 and its relatives. Bar, 0.01 nucleotide substitutions per nucleotide position.

**Figure 3 molecules-28-06738-f003:**
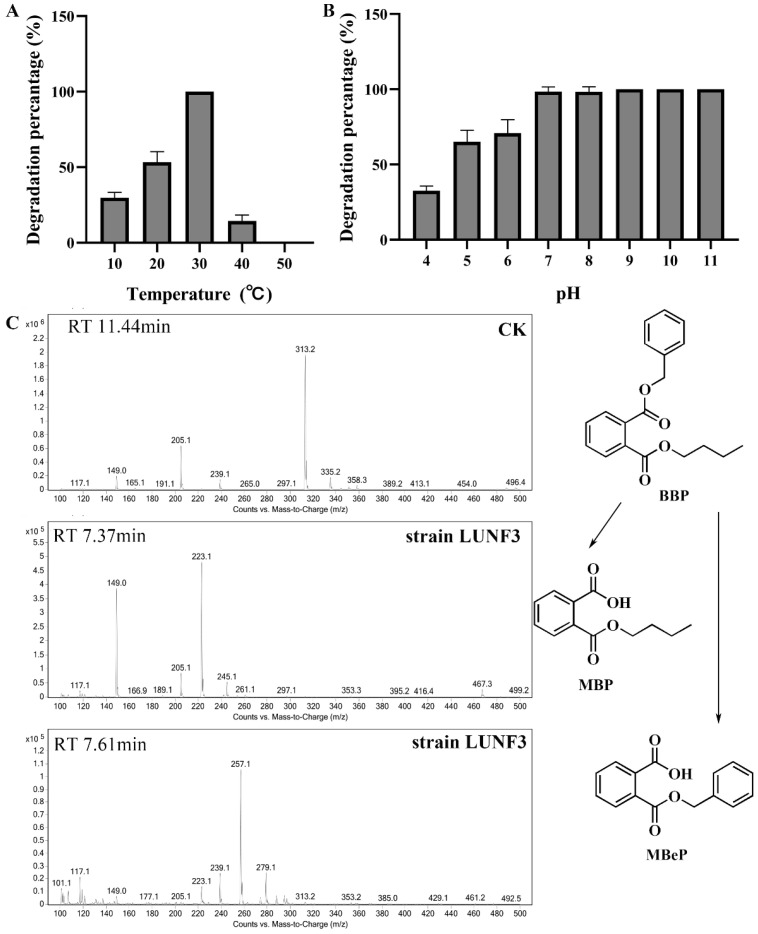
The characterization of strain LUNF3 degrading PAEs. The degradation performance of strain LUNF3 against DBP under a range of temperature (**A**) and pH values (**B**). Strain LUNF3 was cultivated in TEM (pH 8.0) with 0.5 mM DBP for 5 days at 10–50 °C (**A**). Strain LUNF3 was cultivated for 5 days at 30 °C in TEM (pH 4.0–11.0) supplemented with 0.5 mM DBP (**B**). (**C**) The metabolites of BBP detected using HPLC-MS. Strain LUNF3 was cultivated in TEM (pH 8.0) with 0.5 mM BBP for 5 days at 180rpm and 30 °C. CK: TEM (pH 8.0) with 0.5 mM BBP.

**Figure 4 molecules-28-06738-f004:**
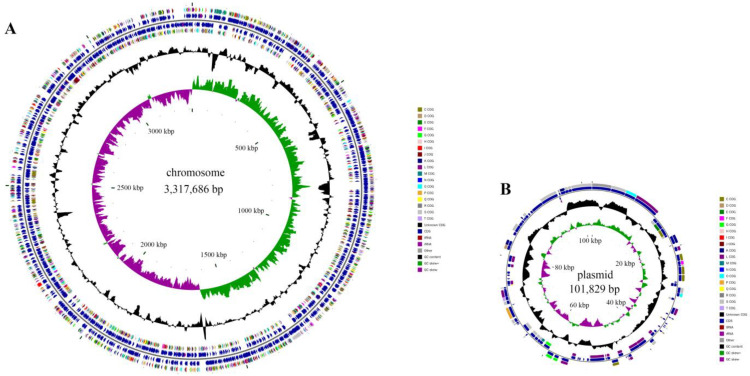
The circle genome map of strain LUNF3 composed of a chromosome (**A**) and a plasmid (**B**). From inside to outside: scale (circle1), GC skew (circle2), GC content (circle3), COG, in which each CDS belonged (circle4 and circle7), and locations of genes (CDS, tRNA and rRNA) in the genome (circle5 and circle6).

**Figure 5 molecules-28-06738-f005:**
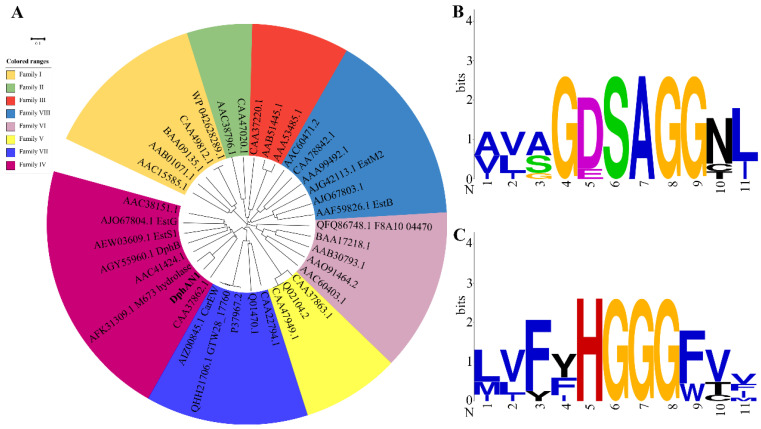
Phylogenetic analysis and conserved sequence analysis of hydrolase DphAN1. (**A**) The phylogenetic tree based on the amino acid sequence of DphAN1 and lipolytic enzymes from families I–VIII. The conserved motifs GDSAGG (**B**) and HGGG (**C**) identified from DphAN1.

**Figure 6 molecules-28-06738-f006:**
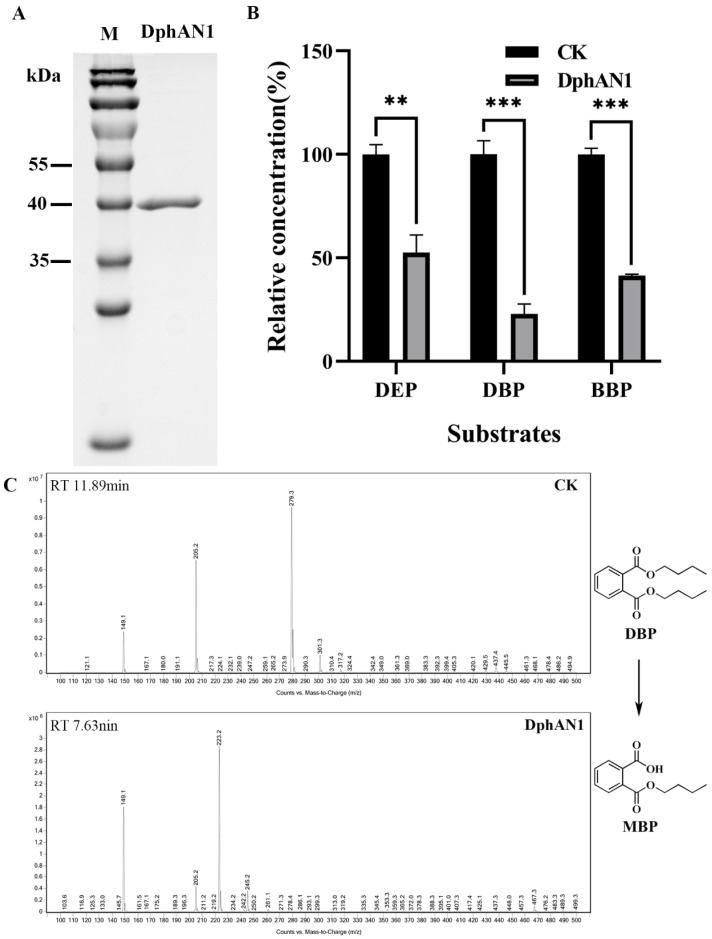
The purification and hydrolysis activity of DphAN1 toward PAEs. (**A**) SDS-PAGE analysis of purified recombinant DphAN1. (**B**) The PAE substrates of DphAN1 (** *p* < 0.01 and *** *p* < 0.0001). In Tris–HCl (50 mM, pH 8.0) supplemented with DEP, DBP or BBP, the hydrolysis reaction by DphAN1 was conducted at 30 °C and 180 rpm for 20 min. CK: Tris–HCl (50 mM, pH 8.0) with 0.5 mM DEP, DBP or BBP. (**C**) HPLC-MS profile of DBP hydrolysis catalyzed by DphAN1. CK: Tris–HCl (50 mM, pH 8.0) with 0.5 mM DBP.

**Figure 7 molecules-28-06738-f007:**
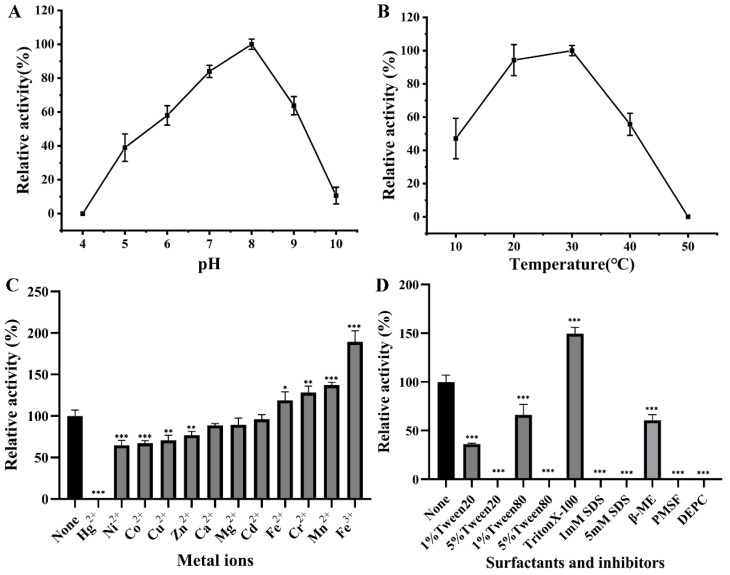
The activity of DphAN1 under various pH values (**A**), temperatures (**B**), metal ions (**C**), surfactants and inhibitors (**D**). In buffer Na_2_HPO_4_–citric acid (pH 4.0–7.0), Tris–HCl (pH 8.0–8.9) and Glycine–NaOH (pH 9.0–10.0) supplemented with DBP, the hydrolysis reaction by DphAN1 was conducted at 30 °C and 180 rpm for 20 min (**A**). DphAN1 and DBP were incubated in Tris–HCl (pH 8.0) at 10–50 °C and 180 rpm for 20 min (**B**). In Na_2_HPO_4_–citric acid (pH 7.0) supplemented with metal ions, DBP hydrolysis reaction by DphAN1 was conducted at 30 °C and 180 rpm for 20 min (**C**) (* *p* < 0.05, ** *p* < 0.01 and *** *p* < 0.0001). In Na_2_HPO_4_–citric acid (pH 7.0) supplemented with surfactants or inhibitors, DBP hydrolysis reaction by DphAN1 was conducted at 30 °C and 180 rpm for 20 min (**D**) (* *p* < 0.05, ** *p* < 0.01 and *** *p* < 0.0001).

**Figure 8 molecules-28-06738-f008:**
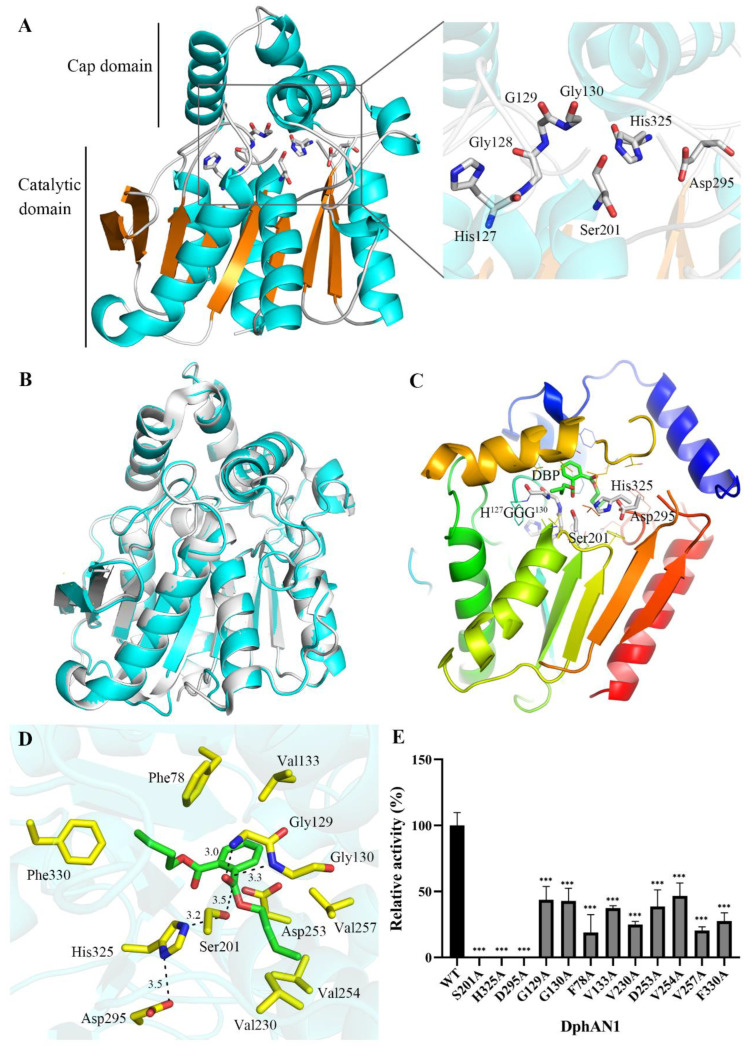
The interaction of DphAN1 and DBP analyzed via homology modeling, molecular docking and site-directed mutagenesis. (**A**) The modeled structure of DphAN1 showing the cap domain and catalytic domain. The catalytic triad (Ser201-Asp295-His325) and oxyanion hole (H127GGG130) are represented as sticks. (**B**) The structural superposition of DphAN1 (cyan) and Est8 (gray). (**C**) The molecular docking of DphAN1 and DBP displaying the substrate-binding pocket between the cap domain and catalytic domain. (**D**) The interaction between amino acid residues of DphAN1 and DBP. (**E**) The relative activity of DphAN1 mutants (*** *p* < 0.0001). In Tris–HCl (pH 8.0) supplemented with DBP, the hydrolysis reaction by DphAN1 or variants was conducted at 30 °C and 180 rpm for 20 min.

**Table 1 molecules-28-06738-t001:** The PAE hydrolases from bacteria.

Hydrolase	Function	Accession No.	Substrate	Hydrolase Family	Origin	Reference
PAE ydrolases	Dialkyl PAE hydrolase	/	DBP, DEHP, DEP, DOP, DMIP, DMTP, DETP	Gene not identified	*Nocardia erythropolis*	[22]
DMP hydrolases	Dialkyl PAE hydrolase	/	DMP	Gene not identified	*Bacillus* sp.	[23]
GTW28_09400	Dialkyl PAE hydrolase	QHH20153.1	DBP, DIBP, DEHP	II	*Bacillus subtilis* BJQ0005	[20]
GTW28_13725	Dialkyl PAE hydrolase	QHH20954.1	DEHP	V	*Bacillus subtilis* BJQ0005	[20]
HylD1	Dialkyl PAE hydrolase	QFQ86055.1	DMP, DEP	IV	*Paracoccus kondratievae* BJQ0001	[26]
HylD2	Dialkyl PAE hydrolase	QFQ86748.1	DEHP	VI	*Paracoccus kondratievae* BJQ0001	[26]
Hyd	Dialkyl PAE hydrolase	AYW76486	DMP, DEP, DBP, DOP, DEHP, BBP, DINP	New family	*Rhodococcus* sp. 2G	[24]
GTW28_17760	Dialkyl/monoalkyl PAE hydrolase	QHH21706.1	DMP, DEP, DBP, DIBP, DEHP, MBP, MEHP	VII	*Bacillus subtilis* BJQ0005	[20]
EstM2	Dialkyl/monoalkyl PAE hydrolase	AJG42113.1	DMP, DEP, BBP, DBP, DPP, MBzP, MMP, MEP, MBP, MPP	VIII	soil metagenomic library	[21]
EstG	Dialkyl/monoalkyl PAE hydrolase	AJO67804.1	DBP	VIII	*Sphingobium* sp. SM42	[19]
MphG1	Monoalkyl PAE hydrolase	AUH70054.1	MEP, MBP, MHP, MEHP	V	*Gordonia* sp. YC-JH1	[18]

DIBP, Diisobutyl phthalate; DINP, Diisononyl phthalate; DPP, Diphenyl phthalate; MEHP, Mono-(2-ethylhexyl) phthalate; MBzP, Monobenzyl phthalate; MMP, Monomethyl phthalate; MEP, Monoethyl phthalate; MBP, Monobutyl phthalate; MPP, Monophenyl phthalate; MHP, Monohexyl phthalate.

## Data Availability

The data in this study can be obtained from the corresponding authors or the NCBI database. The sequences of 16S rRNA gen and *dphAN1* were deposited in GenBank under accession numbers OM900052 and OM908374, respectively. The sequences of chromosome and plasmid were deposited in the GenBank database under accession numbers CP093968 and CP093969, respectively.

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
