# Peer review of "A Novel and Efficient Phthalate Hydrolase from Acinetobacter sp. LUNF3: Molecular Cloning, Characterization and Catalytic Mechanism"

_molecules, 2023, doi:10.3390/molecules28186738_

Round 1
Reviewer 1 Report
The manuscript by Fan et al. is a very complete report, starting with the isolation from contaminated soil of a bacterial strain that hydrolyzes dibutyl phthalate (DBP) to monobutyl phthalate (MBP), as well as degrading other phthalic acid esters (PAEs). The products of PAEs degradation were identified by HPLC-MS. By 16S rRNA sequencing, the isolated bacteria were identified as a strain of Acinetobacter sp. which was named LUNF3. Its genome, composed of a ≈3.3 Mb chromosome and ≈0.1 Mb plasmid, was sequenced, annotated and deposited in GenBank. By genome comparison, a high similitude was observed to the genomes of several Acinetobacter johnsonii strains. A gene encoding putative PAEs hydrolase was identified in the LUNF3 genome by homology to PAEs hydrolases from other bacteria. The LUNF3 gene was named dphAN1 and encodes protein DphAN1, highly identical to a characterized PAEs hydrolase from Acinetobacter sp. M673. Recombinant DphAN1 was expressed from the cloned gene. Highly pure enzyme was characterized as PAEs hydrolase with different substrates, at diverse pH values and temperaturas, and in the presence of metal ions, surfactants and inhibitors. The tridimensional structure of DphAN1 was modeled by homology, and the binding of substrate was simulated by molecular docking of DBP. This allowed the identification of catalytic and substrate-binding residues. The relevance of the residues was tested by site-directed mutagénesis.
As stated, the research described in the manuscript is very complete. However, there are many points that need clarification.
MAJOR POINTS
1.- One of the major problems of the manuscript is that it needs detailed correction by a native speaker, given the frequent use of wrong words or expressions. Some examples are listed below, but the full text should be revised.
Line 19: “hydrolases” should replace “hydrolase”
Line 20 (and lots of similar instances throughout the paper): “PAEs-catalyzing hydrolases“ is wrong because enzymes catalyze reactions, not substrates. (see also lines 71, 95, 220, 316)
Line 30: delete “in”
Line 31: “interaction with” should replace “interacting”
Line 50: “due to their” should replace “with”
Line 51: insert “the” behore “main”
Line 52: “eliminate” should replace “eliminating”
Line 52: “the advantages” should replace “an advantage”
Lines 84-85: the sentence about Hyd hydrolase is difficult to understand
Line 91: “heterologous” should replace “heterogeneous”
Line 92: “characteristics were” should replace “characteristic was”
Line 127: “at” should replace “during”
Line 131: delete “was”
Line 137: “degradative activity” should replace “degradability”
Line 140: “as it is known to occur with”should replace “based on the poor or no degradation by”
Line 147: “minor of strains” is not intelligible
Line 153: “through” should replace “though”
Line 165: “strain” should replace “stain”
Line 177: “according to annotations” should replace “base on annotation”
Line 227: “Environmental factors have” should replace “The environmental factors had”
Line 227: “. So, it was” should replace “, and it was”
Line 237: “will make” should replace “made”
Line 245: “indicating” should replace “indicated”
lines 247-248: delete “. The kinds and concentration of chemicals resulted in discrepant effects on the 247 hydrolytic activity of DphAN1”; it is confuse and useless.
Line 259: “after the” should replace “by”
Line 261: “, thus reducing the” should replace “to lead to reduction in”
Line 263: “Purity” should replace “Purification”
line 349-350: “degradative activity of strain LUNF3” should replace “degradeability of the strain”
Lines 350-351: “estimated by HPLC-MS assay of DBP hydrolysis” should replace “determined according to the reduction of DBP and the metabolites detected by HPLC-MS”
Line 384: “public databases” should replace “pubic database”
Line 389: “with the query of PAEs hydrolases reported” is not intelligible
Line 406: “respectively. 0.5 mM PAEs above and” is not intelligible
Line 418: “DphAN1, these” should replace “DphAN1,These”
Line 424: delete “above”
Line 447: “added with strong aggitation” should replace “violently mixed with the mixture”
Line 463: “adds to known” should replace “developed”
Line 464: “with” should replace ”, and strongly implied its”
Line 468: “provide” should replace “contributed to shed”
Lines 468-469: “may be a basis to improve” should replace “promoted protein structural modification for more excellent”
2.- Lines 31-33 in the abstract are confuse. The “excellebt catalytic performance” cannot contribute to protein structural modification to improve its catalytic efficiency. On the other hand, knowledge of molecular catalytic mechanism may contribute to the design of structural modifications. These lines should be redacted in a different way.
3.- The paragraph describing PAEs hydrolases (lines 63–86) is difficult to follow. A descriptive table listing systematically these enzymes with their basic characteriztics would be very helpful to the reader.
4.- A major argument for the relevance of this reserach is that the catalytic mechanisms of PAEs hydrolases are little studied. According to lines 84-85 it would seem that reference #23 about a hydrolase named Hyd is the only relevant antecedent. However, this is not so. A search in Scholar Google for “phthalate AND hydrolase AND mechanism” yielded at least five relevant references:
A (NOT CITED): Cheng J, Du H, Zhou MS, Ji Y, Xie YQ, Huang HB, Zhang SH, Li F, Xiang L, Cai QY, Li YW, Li H, Li M, Zhao HM, Mo CH. Substrate-enzyme interactions and catalytic mechanism in a novel family VI esterase with dibutyl phthalate-hydrolyzing activity. Environ Int. 2023 Jun 19;178:108054. doi:10.1016/j.envint.2023.108054. Epub ahead of print. PMID: 37354883.
B (NOT CITED): Chen Y, Wang Y, Xu Y, Sun J, Yang L, Feng C, Wang J, Zhou Y, Zhang ZM, Wang Y. Molecular insights into the catalytic mechanism of plasticizer degradation by a monoalkyl phthalate hydrolase. Commun Chem. 2023 Mar 1;6(1):45. doi:10.1038/s42004-023-00846-0. PMID: 36859434; PMCID: PMC9977937.
C REFERENCE #20: Xu Y, Liu X, Zhao J, Huang H, Wu M, Li X, Li W, Sun X, Sun B. An efficient phthalate ester-degrading Bacillus subtilis: Degradation kinetics, metabolic pathway, and catalytic mechanism of the key enzyme. Environ Pollut. 2021 Jan 15;273:116461. doi: 10.1016/j.envpol.2021.116461. Epub ahead of print. PMID: 33485001.
D REFERENCE #23: Du H, Hu RW, Zhao HM, Huang HB, Xiang L, Liu BL, Feng NX, Li H, Li YW, Cai QY, Mo CH. Mechanistic insight into esterase-catalyzed hydrolysis of phthalate esters (PAEs) based on integrated multi-spectroscopic analyses and docking simulation. J Hazard Mater. 2021 Apr 15;408:124901. doi:10.1016/j.jhazmat.2020.124901. Epub 2020 Dec 19. PMID: 33360702.
E REFERENCE #18: Fan S, Wang J, Yan Y, Wang J, Jia Y. Excellent Degradation Performance of a Versatile Phthalic Acid Esters-Degrading Bacterium and Catalytic Mechanism of Monoalkyl Phthalate Hydrolase. Int J Mol Sci. 2018 Sep 18;19(9):2803. doi:10.3390/ijms19092803. PMID: 30231475. PMCID: PMC6164851.
Three of these references are cited in the manuscript as references #18, #20 and #23. As stated, only the latter seems to be considered as a relevant antecedent for mechanistic studies of PAEs hydrolases (lines 84-85), although reference #18 by Fan et al. is from the same first author. On the other hand, the two most recent references found in PubMed (those by Cheng et al. 2023 and Chen et al. 2023) are not cited.
In summary, the question of known antecedents for mechanistic studies of PAEs hydrolases should be reconsidered by the authors and the manuscript modified accordingly, including comparisons with DphAN1.
5.- The meaning of CK in Figures 1A, 2C, 5B, 5C, S1A, S2A, S2B, S7A and S7B, is nuclear. Please, define.
6.- The meaning of the 0.01 scale in the lower part of Figure 1B is nuclear. Please, define.
7.- In Figures 1, 2, 5, 6, 7E, please specify conditions of assays
8.- In lines 181-182, please specificy percent identities of DphAN1 with PAEs hydrolases from Acinetobacter M673 and from Sulphobacillus (separately).
9.- In the NCBI Protein record of DphAN1 (accession number UNW57817.1), the link “Identical proteins” recovers 100% identical proteins from Acinetobacter johnsonii. Please, comment on the significance of this with respect to the current manuscript.
10.- The effects of some metal ions and detergents are described as great(ly) increases (abstract lines 27-28). This is somewhat exagerated. They should be rather considered as moderate increases.
11.- Line 315-317. Please, suggest how the identification of residues involved in DBP binding and hydrolysis would help to improve catalytic efficiency of DphAN1 by rational design.
12.- For the assays reported in Figure 7E, what were the criteria used to standardize the comparison of the different point mutants with the wildtype? This is very important for the reliability of the comparison. In Figure S10, SDS-PAGE of wild type and three mutants is shown. The rest of the mutants should be also shown in this Figure.
13.- Line 383. The Victorian Bioinformatics Consortium (VBC) Closed on 31 DEC 2014. Please, check and specificy date of access to the resource.
14. line 408. The mention to 180 rpm is intriguing. Please correct or explain.
The English needs extensive correction. Many points, but not all, are indicated in the report
Author Response
Thank you for your valuable comments concerning our manuscript entitled “A novel and efficient phthalate hydrolase from Acinetobacter sp. LUNF3: molecular cloning, characterization and catalytic mechanism” (Manuscript ID: molecules-2574942).
Those comments are very helpful for revising and improving our manuscript, as well as the important guidance to our following research.
We have read these comments carefully and made corresponding corrections which we hope to meet with approval.
Revised portion of the manuscript has been highlighted using the "Track Changes" function in Microsoft Word (marked in red). The main corrections corresponding to the comments are as follows.
Point 1: One of the major problems of the manuscript is that it needs detailed correction by a native speaker, given the frequent use of wrong words or expressions. Some examples are listed below, but the full text should be revised.
Response 1: Thank you very much for your suggestion. This manuscript has undergone English language editing by MDPI. The text has been checked for correct use of grammar and common technical terms, and edited to a level suitable for reporting research in a scholarly journal. Some expressions which are not intelligible have been modified.
For example, Line 388-389 (lines 487-490 in revised manuscript) : the sentence contains “with the query of PAEs hydrolases reported” has been modified as “The genome of strain LUNF3 was mined and searched for the similar sequences of PAE hydrolases reported such as XtjR8, HylD1, HylD2 and EstM2 [1-3].”
Line 406 ( line 507 in revised manuscript ): “respectively. 0.5 mM PAEs above and” has been corrected as “respectively. 0.5 mM substrate and”.
Point 2: Lines 31-33 in the abstract are confuse. The “excellent catalytic performance” cannot contribute to protein structural modification to improve its catalytic efficiency. On the other hand, knowledge of molecular catalytic mechanism may contribute to the design of structural modifications. These lines should be redacted in a different way.
Response 2: According to your suggestion, this part has been modified as follows: “The molecular catalytic mechanism of DphAN1 contribute to its structural modification, improving its catalytic efficiency, and allowing for its application in bioremediation.”
Point 3: The paragraph describing PAEs hydrolases (lines 63–86) is difficult to follow. A descriptive table listing systematically these enzymes with their basic characteriztics would be very helpful to the reader.
Response 3: Thank you for your suggestion. Figure 1 and Table 1 have been added in this article. Figure 1 displays the pathways of PAEs hydrolyzed by microorganisms. Table 1 describes these PAE hydrolases with their basic characteristics.
Figure 1. The pathways of PAEs hydrolyzed by microorganisms.
|
Table 1. The PAE hydrolases from bacteria |
||||||
|
Hydrolase |
Function |
Accession No. |
Substrate |
Hydrolase family |
Origin |
Reference |
|
PAEs hydrolase |
Dialkyl PAE hydrolase |
/ |
DBP,DEHP, DEP, DOP, DMIP,DMTP, DETP |
Gene not identified |
Nocardia erythropolis |
[22] |
|
DMP hydrolases |
Dialkyl PAE hydrolase |
/ |
DMP |
Gene not identified |
Bacillus sp. |
[23] |
|
GTW28_09400 |
Dialkyl PAE hydrolase |
QHH20153.1 |
DBP,DIBP, DEHP |
II |
Bacillus subtilis BJQ0005 |
[20] |
|
GTW28_13725 |
Dialkyl PAE hydrolase |
QHH20954.1 |
DEHP |
V |
Bacillus subtilis BJQ0005 |
[20] |
|
HylD1 |
Dialkyl PAE hydrolase |
QFQ86055.1 |
DMP, DEP |
IV |
Paracoccus kondratievae BJQ0001 |
[26] |
|
HylD2 |
Dialkyl PAE hydrolase |
QFQ86748.1 |
DEHP |
VI |
Paracoccus kondratievae BJQ0001 |
[26] |
|
Hyd |
Dialkyl PAE hydrolase |
AYW76486 |
DMP,DEP, DBP,DOP, DEHP, BBP, DINP |
New family |
Rhodococcus sp. 2G |
[24] |
|
GTW28_17760 |
Dialkyl/ monoalkyl PAE hydrolase |
QHH21706.1 |
DMP,DEP, DBP,DIBP, DEHP,MBP, MEHP |
VII |
Bacillus subtilis BJQ0005 |
[20] |
|
EstM2 |
Dialkyl/ monoalkyl PAE hydrolase |
AJG42113.1 |
DMP,DEP, BBP,DBP, DPP, MBzP MMP,MEP, MBP, MPP |
VIII |
soil metagenomic library |
[21] |
|
EstG |
Dialkyl/ monoalkyl PAE hydrolase |
AJO67804.1 |
DBP |
VIII |
Sphingobium sp. SM42 |
[19] |
|
MphG1 |
Monoalkyl PAE hydrolase |
AUH70054.1 |
MEP, MBP, MHP, MEHP |
V |
Gordonia sp. YC-JH1 |
[18] |
DIBP, Diisobutyl phthalate; DINP, Diisononyl phthalate; DPP, Diphenyl phthalate; MEHP, Mono-(2-ethylhexyl) phthalate; MBzP, Monobenzyl phthalate; MMP, Monomethyl phthalate; MEP, Monoethyl phthalate; MBP, Monobutyl phthalate; MPP, Monophenyl phthalate; MHP, Monohexyl phthalate.
Point 4: A major argument for the relevance of this reserach is that the catalytic mechanisms of PAEs hydrolases are little studied. According to lines 84-85 it would seem that reference #23 about a hydrolase named Hyd is the only relevant antecedent. However, this is not so. A search in Scholar Google for “phthalate AND hydrolase AND mechanism” yielded at least five relevant references:
A (NOT CITED): Cheng J, Du H, Zhou MS, Ji Y, Xie YQ, Huang HB, Zhang SH, Li F, Xiang L, Cai QY, Li YW, Li H, Li M, Zhao HM, Mo CH. Substrate-enzyme interactions and catalytic mechanism in a novel family VI esterase with dibutyl phthalate-hydrolyzing activity. Environ Int. 2023 Jun 19;178:108054. doi:10.1016/j.envint.2023.108054. Epub ahead of print. PMID: 37354883.
B (NOT CITED): Chen Y, Wang Y, Xu Y, Sun J, Yang L, Feng C, Wang J, Zhou Y, Zhang ZM, Wang Y. Molecular insights into the catalytic mechanism of plasticizer degradation by a monoalkyl phthalate hydrolase. Commun Chem. 2023 Mar 1;6(1):45. doi:10.1038/s42004-023-00846-0. PMID: 36859434; PMCID: PMC9977937.
C REFERENCE #20: Xu Y, Liu X, Zhao J, Huang H, Wu M, Li X, Li W, Sun X, Sun B. An efficient phthalate ester-degrading Bacillus subtilis: Degradation kinetics, metabolic pathway, and catalytic mechanism of the key enzyme. Environ Pollut. 2021 Jan 15;273:116461. doi: 10.1016/j.envpol.2021.116461. Epub ahead of print. PMID: 33485001.
D REFERENCE #23: Du H, Hu RW, Zhao HM, Huang HB, Xiang L, Liu BL, Feng NX, Li H, Li YW, Cai QY, Mo CH. Mechanistic insight into esterase-catalyzed hydrolysis of phthalate esters (PAEs) based on integrated multi-spectroscopic analyses and docking simulation. J Hazard Mater. 2021 Apr 15;408:124901. doi:10.1016/j.jhazmat.2020.124901. Epub 2020 Dec 19. PMID: 33360702.
E REFERENCE #18: Fan S, Wang J, Yan Y, Wang J, Jia Y. Excellent Degradation Performance of a Versatile Phthalic Acid Esters-Degrading Bacterium and Catalytic Mechanism of Monoalkyl Phthalate Hydrolase. Int J Mol Sci. 2018 Sep 18;19(9):2803. doi:10.3390/ijms19092803. PMID: 30231475. PMCID: PMC6164851.
Three of these references are cited in the manuscript as references #18, #20 and #23. As stated, only the latter seems to be considered as a relevant antecedent for mechanistic studies of PAEs hydrolases (lines 84-85), although reference #18 by Fan et al. is from the same first author. On the other hand, the two most recent references found in PubMed (those by Cheng et al. 2023 and Chen et al. 2023) are not cited.
In summary, the question of known antecedents for mechanistic studies of PAEs hydrolases should be reconsidered by the authors and the manuscript modified accordingly, including comparisons with DphAN1.
Response 4: Thank you very much for your suggestion. The part lines 84-85 has been corrected as follows: “Although the catalytic mechanisms of dialkyl PAE hydrolase PS06828 and Hyd, from family VI and a new family respectively, are proposed by spectroscopic and docking analyses [4,5], only the catalytic triad of PS06828 (Ser113, Asp166, and His197) and key residues of Hyd (Thr190 and Ser191) are revealed. The results of molecular docking and enzyme assay show the interaction between MIBP and His399 affect the activity of GTW28_17760, a hydrolase from family VII capable of hydrolyzing DIBP and MIBP [6]. The catalytic mechanisms of some monoalkyl PAE hydrolases from family V are also elucidated [7,8]. Together, the catalytic mechanisms of more PAE hydrolase from other families requires to be further resolved.”
Point 5: The meaning of CK in Figures 1A, 2C, 5B, 5C, S1A, S2A, S2B, S7A and S7B, is nuclear. Please, define.
Response 5: Thank you for your suggestion. The meaning of CK in Figures 1A, 2C, 5B, 5C, S1A, S2A, S2B, S7A and S7B (Figures 2A, 3C, 6B, 6C, S1A, S2A, S2B, S7A and S7B in revised manuscript) has been defined.
“Figure 2. Functional and molecular identification of strain LUNF3. (A) HPLC-MS profile of DBP degraded by strain LUNF3 with the metabolite of MBP. Strain LUNF3 was cultivated in TEM (pH8.0) with 0.5 mM DBP for 5 days at 180rpm and 30 ºC. CK: TEM (pH8.0) with 0.5 mM DBP. (B) The phylogenetic tree constructed using 16S rRNA gene sequences of strain LUNF3 and its relatives. Bar, 0.01 nucleotide substitutions per nucleotide position.
Figure 3. The characterization of strain LUNF3 degrading PAEs. The degradation performance of strain LUNF3 against DBP under a range of temperature (A) and pH values (B). Strain LUNF3 was cultivated in TEM (pH8.0) with 0.5 mM DBP for 5 days at 10-50 ºC (A). Strain LUNF3 was cultivated for 5 days at 30 ºC in TEM (pH4.0-11.0) supplemented with 0.5 mM DBP (B). (C) The metabolites of BBP detected using HPLC-MS. Strain LUNF3 was cultivated in TEM (pH8.0) with 0.5 mM BBP for 5 days at 180rpm and 30 ºC. CK: TEM (pH8.0) with 0.5 mM BBP.
Figure 6. The purity and hydrolysis activity of DphAN1 toward PAEs. (A) SDS-PAGE analysis of purified recombinant DphAN1. (B) The PAEs substrates of DphAN1 (* p < 0.05, ** p < 0.01 and *** p < 0.0001). In Tris-HCl (50 mM, pH8.0) supplemented with DEP, DBP or BBP, the hydrolysis reaction by DphAN1 was conducted out at 30 °C and 180 rpm for 20 min. CK: Tris-HCl (50 mM, pH8.0) with 0.5 mM DEP, DBP or BBP. (C) HPLC-MS profile of DBP hydrolysis catalyzed by DphAN1. CK: Tris-HCl (50 mM, pH8.0) with 0.5 mM DBP.Figure S1. The HPLC profile of DBP degraded by strain LUNF3 (B) compared with control (A). CK: TEM (pH8.0) with 0.5 mM DBP.
Figure S2. The substrates detection of strain LUNF3 (A) and metabolites detected by HPLC-MS (B) (* p < 0.05, ** p < 0.01 and *** p < 0.0001). CK (A): TEM (Ph8.0) with 0.5 Mm DEP, DBP or BBP. CK (B): TEM (pH8.0) with 0.5 mM DEP.
Figure S7. The HPLC-MS spectrum of DEP (A) and BBP (B) catalyzed by DphAN1. CK (A): Tris-HCl (50 mM, pH8.0) with 0.5 mM DEP. CK (B): Tris-HCl (50 mM, pH8.0) with 0.5 mM BBP.”
Point 6: The meaning of the 0.01 scale in the lower part of Figure 1B is nuclear. Please, define.
Response 6: Thank you for your suggestion. The meaning of the 0.01 scale in the lower part of Figure 1B (Figures 2B in revised manuscript) has been defined.
“Figure 2. Functional and molecular identification of strain LUNF3. (A) HPLC-MS profile of DBP degraded by strain LUNF3 with the metabolite of MBP. Strain LUNF3 was cultivated in TEM (pH8.0) with 0.5 mM DBP for 5 days at 180rpm and 30 ºC. CK: TEM (pH8.0) with 0.5 mM DBP. (B) The phylogenetic tree constructed using 16S rRNA gene sequences of strain LUNF3 and its relatives. Bar, 0.01 nucleotide substitutions per nucleotide position.”
Point 7: In Figures 1, 2, 5, 6, 7E, please specify conditions of assays.
Response 7: Thank you for your suggestion. In Figures 1, 2, 5, 6 and 7E (Figures 2, 3, 6, 7 and 8E in revised manuscript), the conditions of assays have been specified.
“Figure 2. Functional and molecular identification of strain LUNF3. (A) HPLC-MS profile of DBP degraded by strain LUNF3 with the metabolite of MBP. Strain LUNF3 was cultivated in TEM (pH8.0) with 0.5 mM DBP for 5 days at 180rpm and 30 ºC. CK: TEM (pH8.0) with 0.5 mM DBP. (B) The phylogenetic tree constructed using 16S rRNA gene sequences of strain LUNF3 and its relatives. Bar, 0.01 nucleotide substitutions per nucleotide position.
Figure 3. The characterization of strain LUNF3 degrading PAEs. The degradation performance of strain LUNF3 against DBP under a range of temperature (A) and pH values (B). Strain LUNF3 was cultivated in TEM (pH8.0) with 0.5 mM DBP for 5 days at 10-50 ºC (A). Strain LUNF3 was cultivated for 5 days at 30 ºC in TEM (pH4.0-11.0) supplemented with 0.5 mM DBP (B). (C) The metabolites of BBP detected using HPLC-MS. Strain LUNF3 was cultivated in TEM (pH8.0) with 0.5 mM BBP for 5 days at 180rpm and 30 ºC. CK: TEM (pH8.0) with 0.5 mM BBP.
Figure 6. The purification and hydrolysis activity of DphAN1 toward PAEs. (A) SDS-PAGE analysis of purified recombinant DphAN1. (B) The PAEs substrates of DphAN1 (* p < 0.05, ** p < 0.01 and *** p < 0.0001). In Tris-HCl (50 mM, pH8.0) supplemented with DEP, DBP or BBP, the hydrolysis reaction by DphAN1 was conducted out at 30 °C and 180 rpm for 20 min. CK: Tris-HCl (50 mM, pH8.0) with 0.5 mM DEP, DBP or BBP. (C) HPLC-MS profile of DBP hydrolysis catalyzed by DphAN1. CK: Tris-HCl (50 mM, pH8.0) with 0.5 mM DBP.
Figure 7. The activity of DphAN1 under various pH values (A), temperatures (B), metal ions (C), surfactants and inhibitors (D). In buffer Na2HPO4–citric acid (pH4.0–7.0), Tris–HCl (pH8.0–8.9) and Glycine–NaOH (pH9.0–10.0) supplemented with DBP, the hydrolysis reaction by DphAN1 was conducted out at 30 °C and 180 rpm for 20 min (A). DphAN1 and DBP was incubated in Tris-HCl (pH8.0) at 10 ºC - 50 ºC and 180 rpm for 20 min (B). In Na2HPO4-citric acid (pH 7.0) supplemented with metal ions, DBP hydrolysis reaction by DphAN1 was conducted out at 30 °C and 180 rpm for 20 min (C) (* p < 0.05, ** p < 0.01 and *** p < 0.0001). In Na2HPO4-citric acid (pH 7.0) supplemented with surfactants or inhibitors, DBP hydrolysis reaction by DphAN1 was conducted out at 30 °C and 180 rpm for 20 min (D) (* p < 0.05, ** p < 0.01 and *** p < 0.0001).
Figure 8. The interaction of DphAN1 and DBP analyzed via homology modeling, molecular docking and site-directed mutagenesis. (A) The modeled structure of DphAN1 showing the cap domain and catalytic domain. The catalytic triad (Ser201-Asp295-His325) and oxyanion hole (H127GGG130) are represented as stick. (B) The structural superposition of DphAN1 (cyan) and Est8 (grey). (C) The molecular docking of DphAN1 and DBP displaying the substrate binding pocket between the cap domain and catalytic domain. (D) The interaction between amino acid residues of DphAN1 and DBP. (E) The relative activity of DphAN1 mutants (* p < 0.05, ** p < 0.01 and *** p < 0.0001). In Tris–HCl (pH8.0) supplemented with DBP, the hydrolysis reaction by DphAN1 or variants was conducted out at 30 °C and 180 rpm for 20 min.”
Point 8: In lines 181-182, please specificy percent identities of DphAN1 with PAEs hydrolases from Acinetobacter M673 and from Sulphobacillus (separately).
Response 8: In lines 181-182 (lines 252-255 in revised manuscript), percent identities of DphAN1 with PAEs hydrolase from Acinetobacter sp. M673 has been specified.
“Among these 109 genes mentioned above, one gene encodes a hydrolase DphAN1, which shares 67.0% identity with reported PAE hydrolase from Acinetobacter sp. M673 [30].”
Point 9: In the NCBI Protein record of DphAN1 (accession number UNW57817.1), the link “Identical proteins” recovers 100% identical proteins from Acinetobacter johnsonii. Please, comment on the significance of this with respect to the current manuscript.
Response 9: Thank you very much. According to the phylogenetic analysis of 16S rRNA gene sequences, strain LUNF3 is taxonomically close to strains Acinetobacter johnsonii CIP 64.6 and Acinetobacter oryzae B23 (Figure 2B). After genome assembly, according to the ANI values of strain LUNF3 against Acinetobacter johnsonii LXL_C1 (GenBank No. CP031011), Acinetobacter johnsonii M19 (GenBank No. CP037424) and Acinetobacter johnsonii FDAARGOS_1093 (GenBank No. CP068195) (97.75%, 96.17% and 95.85%, respectively), strain LUNF3 can be a member of Acinetobacter johnsonii. Therefore, the genomes of other strains of Acinetobacter johnsonii may have the gene coding DphAN1. However, there have been no reports about the PAE hydrolysis activity of this gene so far. In this work, DphAN1 was identified to be capable of hydrolyze PAEs. So, these results have novelty.
Point 10: The effects of some metal ions and detergents are described as great(ly) increases (abstract lines 27-28). This is somewhat exaggerated. They should be rather considered as moderate increases.
Response 10: Thank you for your suggestion. The description in lines 27-29 has been modified as follows:
“Moreover, the metal ions (Fe2+, Mn2+, Cr2+and Fe3+) and surfactant TritonX-100 significantly activated DphAN1, indicating high adaptability and tolerance of DphAN1 to these chemicals.”
Point 11: Line 315-317. Please, suggest how the identification of residues involved in DBP binding and hydrolysis would help to improve catalytic efficiency of DphAN1 by rational design.
Response 11: Thank you very much. The identification of key residues of hydrolase DphAN1, which are involved in DBP binding and hydrolysis, can provide structural basis for protein engineering. Then the protein structure of DphAN1 can be modified by rational design to optimize catalytic efficiency. Austin et al. resolved X-ray crystal structure of PETase (PET-digesting enzyme) and revealed key residues [9]. Then, two active-site residues were mutated and the binding cleft was narrowed, and the improved PET degradation was observed.
Point 12: For the assays reported in Figure 7E, what were the criteria used to standardize the comparison of the different point mutants with the wildtype? This is very important for the reliability of the comparison. In Figure S10, SDS-PAGE of wild type and three mutants is shown. The rest of the mutants should be also shown in this Figure.
Response 12: Thank you for your suggestion. For the assays reported in Figure 7E (Figure 8E in revised manuscript), The activity of wild type DphAN1 was defined as 100%, and the activities of DphAN1 mutants were expressed as relative activity to that of wild type DphAN1. In Figure S10, the wild type and 13 mutants is shown in the gel.
Point 13: Line 383. The Victorian Bioinformatics Consortium (VBC) Closed on 31 DEC 2014. Please, check and specificy date of access to the resource.
Response 13: Thank you very much. The corresponding part (lines 461-464 in revised manuscript) has been modified as follows: “Based on the complete genome sequence, protein-coding genes, tRNA genes and rRNA genes were predicted by GeneMarkS, Trnascan-se and RNAmmer [66-68], respectively.”
Point 14: line 408. The mention to 180 rpm is intriguing. Please correct or explain.
Response 14: Thank you very much. Phthalic acid esters (PAEs) have low water solubility [10]. Agitation may enhance the contact between hydrolase DphAN1 and DBP, and improve the catalytic efficiency. Lin reported the improved conversion of amidase with the increase of the agitation rate [11]. The optimal conditions for C. antarctica lipase A catalyzed enantioselective hydrolysis of (R,S)-2,3-2-PPAME including agitation speed at 400 rpm [12]. Therefore, Agitation can facilitate the catalytic reaction of lipolytic enzymes.
References
- Qiu, J.; Yang, H.; Yan, Z.; etc. Characterization of XtjR8: A novel esterase with phthalate-hydrolyzing activity from a metagenomic library of lotus pond sludge. Int J Biol Macromol 2020, 164, 1510-1518.
- Xu, Y.; Minhazul, K.A.H.M.; Wang, X.; etc. Biodegradation of phthalate esters by Paracoccus kondratievae BJQ0001 isolated from Jiuqu (Baijiu fermentation starter) and identification of the ester bond hydrolysis enzyme. Environ Pollut 2020, 263, 114506.
- Sarkar, J.; Dutta, A.; Pal Chowdhury, P.; etc. Characterization of a novel family VIII esterase EstM2 from soil metagenome capable of hydrolyzing estrogenic phthalates. Microb Cell Fact 2020, 19.
- Cheng, J.; Du H; Zhou, M.S.; etc. Substrate-enzyme interactions and catalytic mechanism in a novel family VI esterase with dibutyl phthalate-hydrolyzing activity. Environ Int 2023, 178, 108054, doi:10.1016/j.envint.2023.108054.
- Du, H.; Hu, R.; Zhao, H.; etc. Mechanistic insight into esterase-catalyzed hydrolysis of phthalate esters (PAEs) based on integrated multi-spectroscopic analyses and docking simulation. J Hazard Mater 2021, 408, 124901.
- Xu, Y.; Liu, X.; Zhao, J.; etc. An efficient phthalate ester-degrading Bacillus subtilis: Degradation kinetics, metabolic pathway, and catalytic mechanism of the key enzyme. Environ Pollut 2021, 273, 116461.
- Chen, Y.; Wang, Y.; Xu, Y.; etc. Molecular insights into the catalytic mechanism of plasticizer degradation by a monoalkyl phthalate hydrolase. Commun Chem 2023, 6, 45, doi:10.1038/s42004-023-00846-0.
- Fan, S.; Wang, J.; Yan, Y.; etc. Excellent Degradation Performance of a Versatile Phthalic Acid Esters-Degrading Bacterium and Catalytic Mechanism of Monoalkyl Phthalate Hydrolase. International Journal of Molecular Sciences 2018, 19, 2803.
- Austin, H.P.; Allen, M.D.; Donohoe, B.S.; etc. Characterization and engineering of a plastic-degrading aromatic polyesterase. Proceedings of the National Academy of Sciences 2018, 115, E4350-E4357.
- Gao, D.; Wen, Z. Phthalate esters in the environment: A critical review of their occurrence, biodegradation, and removal during wastewater treatment processes. Sci Total Environ 2016, 541, 986-1001.
- Lin, C.P.; Tang, X.L.; Zheng, R.C.; etc. Efficient chemoenzymatic synthesis of (S)-alpha-amino-4-fluorobenzeneacetic acid using immobilized penicillin amidase. Bioorg Chem 2018, 80, 174-179, doi:10.1016/j.bioorg.2018.06.020.
- Cheng, Q.; Liu, G.; Zhang, P.; etc. Lipase-catalyzed hydrolysis of (R,S)-2,3-diphenylpropionic methyl ester enhanced by hydroxypropyl-beta-cyclodextrin. Biotechnol Progr 2018, 34, 1355-1362, doi:10.1002/btpr.2716.

Reviewer 2 Report
Dear Editor,
The MS entitled “A novel and efficient phthalate hydrolase from Acinetobacter sp. LUNF3: molecular cloning, characterization and catalytic mechanism” describes the characterization of PAEs-catalyzing hydrolases from Acinetobacter sp. The authors have done a good piece of work and but needs major revision on the following issues and improvement in English language.
- Introduction lacks clarity at some places. It looks as review of literature. At the end, the rational of the study and clear objective must be stated.
- How the identification of organism was done? What is meant by maximum identity? The author must give the details of percent identity in tabular form.
- How the phylogeny was done? Which software or tool was used?
- What is TEM?
- The sentence should not start with a numerical figure.
- Which tests were applied for statistical analysis?
- Abbreviation must be written when used at the start?
- In result section, legends can be improved.
- In Figure 2, why error bars are not present at 20 and pH 9-11?
- In Figure 3, which tool was used for making genome map? How the reference strains were selected?
- In Figure 6, error bars show difference at what level of significance?
- In Figure 7E, what do the stars indicate?
- Legends for Tables must be improved. There is no legend for Table 4.
- Discussion can be improved by citing the figures and tables and appropriate literature.
There are many mistakes, and language errors in the MS which needs improvement.
Author Response
Thank you for your valuable comments concerning our manuscript entitled “A novel and efficient phthalate hydrolase from Acinetobacter sp. LUNF3: molecular cloning, characterization and catalytic mechanism” (Manuscript ID: molecules-2574942).
Those comments are very helpful for revising and improving our manuscript, as well as the important guidance to our following research.
We have read these comments carefully and made corresponding corrections which we hope to meet with approval.
Revised portion of the manuscript has been highlighted using the "Track Changes" function in Microsoft Word (marked in red). Line has changed since the manuscript has been revised. The main corrections corresponding to the comments are as following:
Point 1: Introduction lacks clarity at some places. It looks as review of literature. At the end, the rational of the study and clear objective must be stated.
Response 1: Thank you for your suggestion. The Introduction part has been extensively improved to be more organized and more clear. The rational and objective of this study were also stated.
Point 2: How the identification of organism was done? What is meant by maximum identity? The author must give the details of percent identity in tabular form.
Response 2: The 16S rRNA gene sequence of strain LUNF3 was submitted in NCBI (https://blast.ncbi.nlm.nih.gov/Blast.cgi?PROGRAM=blastn&PAGE_TYPE=BlastSearch&LINK_LOC=blasthome) to search for the relatives. Strain LUNF3 is closely related to Acinetobacter sp., 16S rRNA gene of strain LUNF3 shares maximum identity with that of Acinetobacter sp. 345 (99.73%) (Table S1 in supplementary material).
Point 3: How the phylogeny was done? Which software or tool was used?
Response 3: The phylogenetic tree of 16S rRNA gene sequences of strain LUNF3 and its relatives was constructed by MEGA 6.0 using neighbor-joining method.
Point 4: What is TEM?
Response 4: The trace element medium (TEM) is kind of inorganic salt medium for bacteria culture [1,2]. TEM is composed of the following components (g/L of distilled water): CaCl2(0.01), K2HPO4 (1.5), (NH4)2SO4 (2.0), MgSO4··7H2O(0.2), Na2HPO4··12H2O (1.5), and trace element solution (TES) (100 μL). TES is composed of the following components (g/L of distilled water): FeSO4··7H2O (5), MnSO4··2H2O (1.43), ZnSO4··7H2O (0.022), CuSO4··5H2O(0.03), Na2WO4··2H2O (0.023), Na2MoO4··2H2O (0.02), and CoSO4··7H2O (0.12).
Point 5: The sentence should not start with a numerical figure.
Response 5: Thank you for your suggestion. We have made corresponding corrections in this manuscript.
Point 6: Which tests were applied for statistical analysis?
Response 6: Thank you very much. In this manuscript t test was applied for statistical analysis.
Point 7: Abbreviation must be written when used at the start?
Response 7: Thank you for your suggestion. We have written the abbreviation at the start in this manuscript.
Point 8: In result section, legends can be improved.
Response 8: Thank you for your suggestion. We have improved the legends in result section.
Point 9: In Figure 2, why error bars are not present at 20 and pH 9-11?
Response 9: Thank you very much. The experiments were conducted in triplicate. The degradation percentage of three repeats for pH 9-11 was 100%, so the error bars was 0.
Point 10: In Figure 3, which tool was used for making genome map? How the reference strains were selected?
Response 10: The Pacbio sequencing data were assembled using HGAP and CANU software to obtain contigs [3,4]. These contigs were corrected by Illumina NovaSeq sequencing data using pilon software. Then the complete genome sequence was obtained and cgview was used to make genome map [5]. The reference strains were selected from National Center for Biotechnology Information (NCBI) based on genome sequence similarity.
Point 11: In Figure 6, error bars show difference at what level of significance?
Response 11: Thank you very much. In Figure 6, error bars show difference at extreme significance.
Point 12: In Figure 7E, what do the stars indicate?
Response 12: Thank you very much. Figure 7E (Figure 8E in revised manuscript) shows the activity of DphAN1 and its variants, and the stars indicate the significant difference of the activity between the variants and DphAN1 (* p < 0.05, ** p < 0.01 and *** p < 0.0001).
Point 13: Legends for Tables must be improved. There is no legend for Table 4.
Response 13: Thank you very much. The legends for tables have been improved or supplemented.
Point 14: Discussion can be improved by citing the figures and tables and appropriate literature.
Response 14: Thank you very much. Discussion have been improved according to your suggestion.
References
- Nahurira, R.; Ren, L.; Song, J.; etc. Degradation of Di(2-Ethylhexyl) Phthalate by a Novel Gordonia alkanivorans Strain YC-RL2. Curr Microbiol 2017, 74, 309-319.
- Ren, L.; Jia, Y.; Ruth, N.; etc. Biodegradation of phthalic acid esters by a newly isolated Mycobacterium sp. YC-RL4 and the bioprocess with environmental samples. Environ Sci Pollut R 2016, 23, 16609-16619.
- Koren, S.; Walenz, B.P.; Berlin, K.; etc. Canu: scalable and accurate long-read assembly via adaptive k-mer weighting and repeat separation. Genome Res 2017, 27, 722-736, doi:10.1101/gr.215087.116.
- Chin, C.; Peluso, P.; Sedlazeck, F.; etc. Phased diploid genome assembly with single-molecule real-time sequencing. Nat Methods 2016, 13.
- Stothard, P.; Wishart, D.S. Circular genome visualization and exploration using CGView. Bioinformatics 2005, 21, 537-539, doi:10.1093/bioinformatics/bti054.

Round 2
Reviewer 1 Report
The English language of the revised version of the manuscript has been thoroughly edited and it is much improved. However, some problems are still to be edited possibly due in part to the lack of scientific knowledge of the english editor. Line numbers indicated below correspond to the tracked-changes manuscript.
1.- Line 27: Abbreviations should be defined in the abstract.
2.- Lines 33-35: This sentence continues to be inexact. It must be substituted by the last sentence of the Conclusions section.
3.- Line 41: delete “since”.
4.- Line 78: delete the comma after “hydrolase”. 5.- Line 95: change “are proposed by” to “had been inferred from”. Line 98: “…show the interaction between MIBP and His399 affect the activity of…” This is not intelligible. Redact in another way. Line 101: change “Together” to “Altogether”. Line 102: change “more PAEs hydrolase” to “PAE hydrolases”. Line 105: Change Figure 1 title to “The pathway of PAE hydrolysis by microorganisms”. Line 144: Change “hydrolase” to “hydrolase(s)” Line 155: Insert “will” before “support”. Change “which lead” to “aiming”. Line 193: delete “that was”. Lines 204-205: Change “although the bulky side chain of BBP might may cause steric hindrance and inhibit hydrolysis as” to “despite the potential steric hindrance by its bulky side chain, “ Figure 4: the insert legends are illegible; the size and resolution of the figure should be increased. Line 290: delete “it was found that”. Line 310: change “remaines” to “remains”. Lines 320-321: the hypothesis of electron transfer in the hydrolytic reaction should be explained or deleted. Line 334: PMSF and DEPC should be defined. Line 380: add “hydrolysis” after DBP. Line 406: insert “were” before “mutated”. Line 477: delete “respectively”. Line 507: delete “respectively”. Replace “Next” with “For the assays”. Line 541: replace “amplifying” with “amplify”. Line 554: replace “detection” with ”chromatographic. Line 570: insert “was” before “validated”. Table S2: change “Lenrth” to “Length” Figure S5 title: change “were” to “are”; and “was” to “is”.
The English is much improved. However, problems are still to be edited. Part of them, but not all are possibly due to the lack of scientific knowledge of the english editor.
Author Response
Thank you for your valuable comments concerning our manuscript entitled “A novel and efficient phthalate hydrolase from Acinetobacter sp. LUNF3: molecular cloning, characterization and catalytic mechanism” (Manuscript ID: molecules-2574942).
Those comments are very helpful for revising and improving our manuscript, as well as the important guidance to our following research.
We have read these comments carefully and made corresponding corrections which we hope to meet with approval.
Revised portion of the manuscript has been highlighted using the "Track Changes" function in Microsoft Word (marked in red). Lines have changed since the manuscript has been revised. The main corrections corresponding to the comments are as following:
Point 1: Line 27: Abbreviations should be defined in the abstract.
Response 1: Thank you for your suggestion. In line 27 (lines 25-26 in revised manuscript), abbreviations DEP, DBP and BBP in the Abstract have been defined as follows: “DphAN1 can hydrolyze DEP (diethyl phthalate), DBP (dibutyl phthalate) and BBP (benzyl butyl phthalate)”.
Point 2: Lines 33-35: This sentence continues to be inexact. It must be substituted by the last sentence of the Conclusions section.
Response 2: Thank you for your suggestion. In lines 33-35(lines 32-33 in revised manuscript), this sentence has been substituted by “These results shed lights into the catalytic mechanism of DphAN1, and may contribute to protein structural modification to improve catalytic efficiency in environment remediation” (lines 33-35 in revised manuscript).
Point 3: Line 41: delete “since”.
Response 3: I am sorry I forgot to delete this word. In Line 41, “since” has been deleted.
Point 4: Line 78: delete the comma after “hydrolase”.
Response 4: Thank you for your suggestion. In line 78 (line 75 in revised manuscript), the comma after “hydrolase” has been deleted.
Point 5: Line 95: change “are proposed by” to “had been inferred from”.
Response 5: Thank you for your suggestion. In line 95 (lines 89-90 in revised manuscript), “are proposed by” has been changed to “have been inferred from”.
Point 6: Line 98: “…show the interaction between MIBP and His399 affect the activity of…” This is not intelligible. Redact in another way.
Response 6: Thank you for your suggestion. In line 98 (lines 92-94 in revised manuscript), “…show the interaction between MIBP and His399 affect the activity of…” has been modified as “ According to the results of molecular docking and enzyme assay show, the inter-action between MIBP and His399 affect the activity of GTW28_17760”.
Point 7: Line 101: change “Together” to “Altogether”.
Response 7: Thank you for your suggestion. In line 101 (line 96 in revised manuscript), “Together” has been changed to “Altogether”.
Point 8: Line 102: change “more PAEs hydrolase” to “PAE hydrolases”.
Response 8: Thank you for your suggestion. In line 102 (line 96 in revised manuscript), “more PAEs hydrolase” has been changed to “PAE hydrolases”.
Point 9: Line 105: Change Figure 1 title to “The pathway of PAE hydrolysis by microorganisms”.
Response 9: Thank you for your suggestion. In line 105 (line 99 in revised manuscript), Figure 1 title has been changed to “The pathway of PAE hydrolysis by microorganisms”.
Point 10: Line 144: Change “hydrolase” to “hydrolase(s)”
Response 10: Thank you for your suggestion. In lines 144-145 (lines 138-139 in revised manuscript), “identify efficient PAE hydrolase and elucidate its molecular catalytic mechanism” has been changed to “identify efficient PAE hydrolases and elucidate their molecular catalytic mechanisms”.
Point 11: Line 155: Insert “will” before “support”. Change “which lead” to “aiming”.
Response 11: Thank you for your suggestion. In line 155 (lines 147-148 in revised manuscript), the word “will” has been inserted before “support”, and “which lead” has been changed to “aiming”.
Point 12: Line 193: delete “that was”.
Response 12: Thank you for your suggestion. In line 193 (line 183 in revised manuscript), “that was” has been deleted.
Point 13: Lines 204-205: Change “although the bulky side chain of BBP might may cause steric hindrance and inhibit hydrolysis as” to “despite the potential steric hindrance by its bulky side chain,
Response 13: Thank you for your suggestion. In lines 204-205 (lines 193-195 in revised manuscript), “although the bulky side chain of BBP might may cause steric hindrance and inhibit hydrolysis as” has been changed to “despite the potential steric hindrance by its bulky side chain”.
Point 14: “ Figure 4: the insert legends are illegible; the size and resolution of the figure should be increased.
Response 14: Thank you very much. The legends of Figure 4 have been modified as: “The circle genome map of strain LUNF3 composed of a chromosome (A) and a plasmid (B). From inside to outside: scale (circle1), GC skew (circle2), GC content (circle3), COG which each CDS belonged to (circle4 and circle7), and locations of genes (CDS, tRNA and rRNA) in the genome (circle5 and circle6).”. The size and resolution of the Figure 4 have been improved.
Point 15: Line 290: delete “it was found that”.
Response 15: Thank you for your suggestion. In line 290 (line 275 in revised manuscript), “it was found that” has been deleted.
Point 16: Line 310: change “remaines” to “remains”.
Response 16: Thank you very much. In line 310 (lines 293-294 in revised manuscript), “remaines” has changed to “remains”.
Point 17: Lines 320-321: the hypothesis of electron transfer in the hydrolytic reaction should be explained or deleted.
Response 17: Thank you for your suggestion. In lines 320-321 (lines 303-304 in revised manuscript), “implying that these metal ions may promote electron transfer during the hydrolysis of DBP” has been deleted.
Point 18: Line 334: PMSF and DEPC should be defined.
Response 18: Thank you for your suggestion. In line 334 (line 317 in revised manuscript), “PMSF and DEPC” have been defined as follows: “Phenyl methane sulfonyl fluoride (PMSF) and diethylpyrocarbonate (DEPC)”.
Point 19: Line 380: add “hydrolysis” after DBP.
Response 19: Thank you for your suggestion. In line 380 (lines 359-360 in revised manuscript), “hydrolysis” has been added after DBP.
Point 20: Line 406: insert “were” before “mutated”.
Response 20: Thank you for your suggestion. In line 406 (line 385 in revised manuscript), “were” has been added before “mutated”.
Point 21: Line 477: delete “respectively”.
Response 21: Thank you for your suggestion. In line 477 (line 452 in revised manuscript), “respectively” has been deleted.
Point 22: Line 507: delete “respectively”. Replace “Next” with “For the assays”.
Response 22: Thank you for your suggestion. In line 507 (line 480 in revised manuscript), “respectively” has been deleted. “Next” has been replaced with “For the assays”.
Point 23: Line 541: replace “amplifying” with “amplify”.
Response 23: Thank you for your suggestion. In line 541 (line 513 in revised manuscript), “amplifying” has been replaced with “amplify”.
Point 24: Line 554: replace “detection” with ”chromatographic.
Response 24: Thank you for your suggestion. In line 554 (line 525 in revised manuscript), “detection” has been replaced with “chromatographic”.
Point 25: Line 570: insert “was” before “validated”.
Response 25: Thank you for your suggestion. In line 570 (line 541 in revised manuscript), “was” has been inserted before “validated”.
Point 26: Table S2: change “Lenrth” to “Length”
Response 26: Thank you for your suggestion. In Table S2, “Lenrth” has been changed to “Length”.
Point 27: Figure S5 title: change “were” to “are”; and “was” to “is”.
Response 27: Thank you for your suggestion. In Figure S5, “were” has been changed to “are”, and “was” has been changed to “is”.

Reviewer 2 Report
Dear Editor,
The manuscript has been revised and necessary changes have been done. In point no. 11, the authors were asked about the level of significance in Figure 6. However, the authors have responded as "error bars show difference at extreme significance" which is not accepted statistically. Rest is fine. The Editor may take clarification on the point as per their own opinion.
Moderate changes may be required
Author Response
Thank you for your valuable comments concerning our manuscript entitled “A novel and efficient phthalate hydrolase from Acinetobacter sp. LUNF3: molecular cloning, characterization and catalytic mechanism” (Manuscript ID: molecules-2574942).
Those comments are very helpful for revising and improving our manuscript, as well as the important guidance to our following research.
We have read these comments carefully and made corresponding corrections which we hope to meet with approval.
Revised portion of the manuscript has been highlighted using the "Track Changes" function in Microsoft Word (marked in red). The main corrections corresponding to the comments are as follows.
Point 1: The manuscript has been revised and necessary changes have been done. In point no. 11, the authors were asked about the level of significance in Figure 6. However, the authors have responded as "error bars show difference at extreme significance" which is not accepted statistically. Rest is fine. The Editor may take clarification on the point as per their own opinion.
Response 1: Thank you very much. In Figure 6, the level of significant difference is shown by p values. p < 0.05 indicates significant difference (*), p < 0.01 indicates extremely significant difference (**), and p < 0.0001 indicates more significant difference (***) than that corresponding to p < 0.01.
